# Explaining the Uncertain:
# Stochastic Shapley Values for Gaussian Process Models

**Siu Lun Chau**
CISPA
Helmholtz Center for Information Security
66123 Saarbrücken, Germany
`siu-lun.chau@cispa.de`

**Krikamol Muandet**[*]
CISPA
Helmholtz Center for Information Security
66123 Saarbrücken, Germany
`muandet@cispa.de`

**Dino Sejdinovic**[*]
School of Computer and Mathematical Sciences
University of Adelaide
Adelaide, South Australia, 5005 Australia
`dino.sejdinovic@adelaide.edu.au`

## Abstract

We present a novel approach for explaining Gaussian processes (GPs) that can utilize the full analytical covariance structure present in GPs. Our method is based on the popular solution concept of Shapley values extended to stochastic cooperative games, resulting in explanations that are random variables. The GP explanations generated using our approach satisfy similar favorable axioms to standard Shapley values and possess a tractable covariance function across features and data observations. This covariance allows for quantifying explanation uncertainties and studying the statistical dependencies between explanations. We further extend our framework to the problem of *predictive explanation*, and propose a *Shapley prior* over the explanation function to predict Shapley values for new data based on previously computed ones. Our illustrations demonstrate the effectiveness of the proposed approach.

## 1   Introduction

Shapley values [1], a solution concept derived from cooperative game theory, are an essential tool in explainable AI that helps understand and interpret the prediction of complex machine learning models. In particular, **SH**apley-**A**dditive-ex**P**lanation (SHAP) [2] algorithms have been applied in various fields, including finance [3], healthcare [4], and robotics [5]. SHAP algorithms compute Shapley values by formulating cooperative game payoffs based on the function obtained from the learning algorithm, e.g., a regressor. This function is often treated as fixed, resulting in deterministic game payoffs. In contrast, we argue that uncertainty in model predictions also plays an equally important role for trustworthy machine learning models as it enables users to make more informed decisions [6], assess the model's confidence [7], and identify areas where more data or improved features are needed [8]. Since uncertainty in model prediction can propagate to downstream explanation, explainability tools should also help users better understand the model's confidence or lack thereof in its predictions, allowing them to calibrate their trust in the explanations. By providing transparency into both the model's predictions and uncertainty, AI systems can become more reliable, interpretable, and trustworthy, facilitating broader adoption across domains where trust is imperative.

---

[*]Equal contribution.

37th Conference on Neural Information Processing Systems (NeurIPS 2023).

Gaussian processes (GPs) [9] are a natural model class for developing an explanation method that can account for predictive uncertainty. While there exist model-specific explanation methods for popular models such as LinearSHAP [10] for linear models, TreeSHAP [11] for trees, DeepSHAP [2] for deep networks, and RKHS-SHAP [12] for kernel methods, there is a lack of GP-specific Shapley value based explanation methods despite the ubiquity of GPs in machine learning. While feature importance can be studied via automatic relevance determination (ARD) lengthscales [9] of the covariance kernel, they only reveal global notions of importance which are not associated to the specific predictions, and are limited to the specific classes of kernels. Yoshikawa and Iwata [13] have proposed using a GP with a local linear regression component for interpretability, but this approach results in a specific interpretable GP model rather than a general GP explanation algorithm. Recently, Marx et al. [14] proposed to sample multiple realisations of the GP function and to apply a model-agnostic explanation algorithm to each sample, thus getting a distribution over explanations. While this in principle can give uncertainties in explanations, it is computationally expensive and not tailored for GPs. It does not take advantage of the various important properties that GPs enjoy, such as the fact that posterior means are reproducing kernel Hilbert space (RKHS) [15] functions and that we have a fully tractable covariance structure. These properties, as we later demonstrate, can lead to more effective estimation of Shapley values and help to understand the statistical dependencies across these explanations. Additionally, this covariance structure allows us to formulate explanations themselves as GPs and thus are able to predict explanations of unseen data.

To address this gap, we present the GP-SHAP algorithm. In Section 2, we contextualize our work by introducing the concepts of stochastic cooperative games and stochastic Shapley values. Both are studied thoroughly by game theorists [16, 17], but have not been utilized by the explanation community. Section 3 characterizes the stochastic cooperative game induced by GPs through the use of the conditional mean process [18, 8], a GP of conditional expectations. We use the weighted least squares formulation of Shapley values [19, 2] to express explanations as multivariate Gaussians with a tractable covariance structure. We then introduce the GP-SHAP algorithm as a method for estimating these stochastic Shapley values. In addition, we extend the GP-SHAP algorithm by combining it with the Bayesian weighted least squares approach by Slack et al. [20]. The extension we call BayesGP-SHAP integrates two sources of uncertainty: GP predictive posterior uncertainty as well as the Shapley value estimation uncertainty (in the cases where Shapley values are too expensive to compute exactly). Section 4 expands beyond GP explanations, demonstrating the applicability of our framework to other predictive models. We consider the setting of predictive explanations and propose a *Shapley prior* to represent explanations for more general predictive models as GPs. This allows us to predict explanations for new data without relying on the standard Shapley value procedure and provide predictive uncertainty to more general explanations beyond GP models, such as explanations from TreeSHAP, DeepSHAP, and KernelSHAP. Illustrations of the effectiveness of GP-SHAP, BayesGP-SHAP, and the Shapley prior for predictive explanations are provided in Section 5 and we conclude the paper in Section 6. All proofs in this paper are given in the appendix.

## 2   Stochastic cooperative games and their Shapley values

In this section, we review the concepts of stochastic cooperative games, denoted as s-games, and their corresponding stochastic Shapley values, referred to as SSVs. These concepts provide the necessary language to introduce GP-SHAP in the coming section. It is important to note that game theorists have studied s-games in various forms [16, 17, 21], but they have not been adequately introduced in the explanation literature. While Covert and Lee [22] briefly discussed s-games, they focused on computing deterministic Shapley values (DSVs) of the mean of the s-game, overlooking the inherent uncertainty in s-games and their potential application to uncertainty-aware explanations.

**Problem formulation.**   The results presented below are analogous to the original work of Shapley's [1], but adapted to games with random outcomes. Formally, let $\Omega$ denote the player set and let $\nu : 2^{\Omega} \to \mathcal{L}_2(\mathbb{R})$ be a s-game with random variable payoff. We restrict our attention to stochastic payoffs with finite second moments as we aim to characterise the variance of the corresponding stochastic Shapley values. However, this is not a necessary condition to show existence and uniqueness of the SSVs. We further require $\nu(\emptyset) = \delta_{\nu_0}$ where $\delta_{\nu_0}$ is the Dirac measure at some constant $\nu_0 \in \mathbb{R}$. We introduce the following concepts to formalise the axioms for SSVs:

**Definition 1** (Carrier). *A carrier of $\nu$ is any $N \subseteq \Omega$ such that $\nu(S) = \nu(N \cap S)$ for all $S \subseteq \Omega$.*

**Definition 2** (Permutation s-game). *Denote $\Pi(\Omega)$ the set of permutations on $\Omega$. For $\pi \in \Pi(\Omega)$, the induced permutation s-game is $\nu_\pi(\pi S) := \nu(S)$ for all $S \subseteq \Omega$, where $\pi S$ is the image of $S$ under $\pi$.*

**Definition 3** (Stochastic value allocation). *Denote $\mathcal{G}$ the space of s-games for $\Omega$. We say $\phi : \mathcal{G} \to \mathcal{L}_2(\mathbb{R})^{|\Omega|}$ is a value allocation of $\nu$ that assigns to each player $i \in \Omega$ a stochastic value $\phi_i(\nu)$.*

The natural extension of the original axioms by Shapley are extended to s-games as follows:

1. **(s-symmetry)** For any $\pi \in \Pi(\Omega), \phi_{\pi i}(\nu_\pi) = \phi_i(\nu)$.
2. **(s-efficiency)** For each carrier $N$ of $\nu$, $\sum_{i \in N} \phi_i(\nu) = \nu(N)$.
3. **(s-linearity)** For any $\nu, \omega \in \mathcal{G}$, $\phi(\nu + \omega) = \phi(\nu) + \phi(\omega)$ where addition of s-games are defined as $(\nu + \omega)(S) := \nu(S) + \omega(S)$ for all $S \subseteq \Omega$.

Note that all equality happens at the random variable level. Unsurprisingly, there is a unique stochastic value allocation that satisfies these three axioms:

**Theorem 4** (Stochastic Shapley values). *The only stochastic value allocation $\phi$ of $\nu$ satisfying s-symmetry, s-efficiency, and s-linearity takes the following form,*

$$\phi_i(\nu) = \sum_{S \subseteq N \setminus \{i\}} c_{|S|} \left( \nu(S \cup i) - \nu(S) \right) \tag{1}$$

*where $N$ is the smallest carrier set of $\Omega$, $c_{|S|} = \frac{1}{|N|} \binom{|N|-1}{|S|}^{-1}$ and $\phi_i(\nu)$ is the $i^{th}$ SSV of s-game $\nu$.*

Note that Equation (1) is equivalent to the DSVs, except it is written in terms of random variables. We emphasise that this result has been proven in Ma et al. [16] using a top-down approach, i.e. starting with Equation (1) and verify it satisfies the stochastic axioms and uniqueness. In the appendix, we offer a contrasting perspective where we mirror Shapley's original bottom-up derivation, i.e. began with the stochastic axioms and subsequently determined the unique solution.

## 2.1 Variances of stochastic Shapley values are not Shapley values of variance games

As (1) is now defined by summing over a weighted differences between random variables, we can analyse the corresponding mean and variances across the SSVs. In the following, denote $\bar{\phi} : \bar{\mathcal{G}} \to \mathbb{R}^{|\Omega|}$ as the deterministic Shapley value allocation where $\bar{\mathcal{G}} := \{\bar{\nu} : \Omega \to \mathbb{R}\}$ is the space of deterministic cooperative games, referred to as d-games from now on.

**Proposition 5.** *Given the player set $\Omega$, let $\nu$ be a stochastic game, $\phi$ a stochastic Shapley value allocation, and $\bar{\phi}$ a deterministic Shapley value allocation. Suppose that $\mathbb{E}[\nu]$ and $\mathbb{V}[\nu]$ are the corresponding mean and variance d-games, respectively. Then, $\mathbb{E}[\phi(\nu)] = \bar{\phi}(\mathbb{E}[\nu])$, but $\mathbb{V}[\phi(\nu)] \neq \bar{\phi}(\mathbb{V}[\nu])$. In particular, the SSV variance is given by*

$$\mathbb{V}[\phi_i(\nu)] = \sum_{S \subseteq N \setminus \{i\}} \sum_{S' \subseteq N \setminus \{i\}} c_{|S|} c_{|S'|} \left( \mathbb{C}[\nu_{S \cup i}, \nu_{S' \cup i}] - \mathbb{C}[\nu_{S \cup i}, \nu_{S'}] - \mathbb{C}[\nu_S, \nu_{S' \cup i}] + \mathbb{C}[\nu_S, \nu_{S'}] \right),$$

*where $\nu_S = \nu(S)$ and $\mathbb{C}$ is the covariance function between the stochastic payoffs.*

Proposition 5 highlights the difference between variances of stochastic Shapley values, which capture the propagated uncertainties through the s-game, and deterministic Shapley values of variance games [23, 24], i.e. deterministic game $\bar{\nu}(S) = \mathbb{V}[\nu(S)]$ for all $S \subseteq \Omega$ for some stochastic game $\nu$. Therefore, in the context of model explanations, variance-based allocation techniques such as the Shapley effects [25] and the work of Fryer et al. [26], where they computed Shapley values on games constructed using the variance and coefficient of determination $R^2$ respectively, cannot be used to capture uncertainty of explanations, as those approaches are themselves explaining the model variance instead. Besides stochastic Shapley values, it is possible to leverage the framework of coalition interval games [27] and interval Shapley values [28] to provide confidence intervals for feature explanations, as Napolitano et al. [29] demonstrated. However, their approach is based on confidence interval of predictive models, thus this frequentist approach is not applicable to Bayesian models such as GPs and the obtained uncertainty around explanations have very different interpretations as well.

At first glance, computing the variance of each stochastic Shapley value seems cumbersome as it requires summing over all possible coalitions twice. However, there exists a compact expression for the full covariance matrix of stochastic Shapley values across players for stochastic games constructed using a Gaussian process model, which we demonstrate in the following section.

# 3 Explaining GPs with GP-SHAP

This section is dedicated to introducing GP-SHAP, where the key idea is about efficiently computing stochastic Shapley values for stochastic games that models conditional expectations of Gaussian processes. Specifically, in Section 3.1, we review the process of constructing Shapley values for model explanations using deterministic cooperative games. We then proceed to formalize the stochastic cooperative game induced by pushing the GP functions through the d-game and demonstrate that the resulting stochastic Shapley values follow a multivariate Gaussian distribution. Section 3.2 is dedicated to the estimation procedure and the introduction of two algorithms for computing GP explanations: GP-SHAP and BayesGP-SHAP. GP-SHAP provides explanations that incorporate the GP predictive uncertainty, while BayesGP-SHAP extends this further by accounting for the additional estimation uncertainty arising in the weighted least squares procedure. This extension is similar to how BayesSHAP introduced by Slack et al. [20] extends SHAP.

## 3.1 Formulating stochastic Shapley values for GP models

Let $X, Y$ be random variables taking values in the $d$-dimensional instance space $\mathcal{X} \subseteq \mathbb{R}^d$ and label space $\mathcal{Y}$ (could be in $\mathbb{R}$ or discrete) respectively with the joint distribution $p(X, Y)$. We use $[d] := \{1, ..., d\}$ to denote feature index set and $S \subseteq [d]$ as the feature index subset. For supervised learning with GPs, the usual goal is to model $P(Y \mid X = x) = \psi(f(\mathbf{x}))$ [9] where $f$ is the GP function of interest and $\psi$ is a problem-specific transformation, e.g., $\psi$ is the identity map for regression and sigmoid transformation for classification problem. We then posit a prior $f \sim \mathcal{GP}(0, k)$ where $k : \mathcal{X} \times \mathcal{X} \to \mathbb{R}$ is a covariance kernel, and compute the posterior distribution (or its approximation) $p(f \mid \mathbf{D})$, which is typically also a GP with posterior mean $\tilde{m}$ and kernel $\tilde{k}$.

**Explaining deterministic $f$ with d-game.** To explain a deterministic function $f$, we create a d-game by treating each feature as a player and constructing the game with $f$. The resulting DSVs then provide feature attributions that satisfy various favourable properties, such as the sum of the attribution equals to the prediction made at that specific point, identical features receive the same attribution, and null feature receives zero attribution [2]. The d-game for local attribution on an observation $\mathbf{x}$ typically involves computing the expected value of $f(X)$ using a reference distribution $r(X \mid X_S = \mathbf{x}_S)$ for different feature subsets $S \subseteq [d]$,

$$\bar{\nu}_f(\mathbf{x}, S) := \mathbb{E}_r[f(X) \mid X_S = \mathbf{x}_S]. \tag{2}$$

This approach, also known as removal-based explanation [30], has been used in various explanation algorithms [2, 11, 12, 31]. These d-games determine the "worth" of features in $S$ by looking at the expectation after "removing" the contribution of other features in $[d] \backslash S$ through integration. The choice of reference distribution leads to explanations with different properties, such as improved locality of estimation [32], promotion of fairness [33], or the incorporation of causal knowledge [34, 35]. In this paper, we focus on the scenario where $r$ corresponds to the data distribution, i.e., $r(X \mid X_S) = p(X \mid X_S)$, as it is one of the most frequently used approaches in the literature [36, 37, 12, 31]. Although different reference functions result in different estimation procedures, our formulation of s-games and SSVs for GPs still holds.

**Induced s-game by stochastic $f$.** Now suppose $f$ is a random function with a posterior distribution $p(f \mid \mathbf{D})$. The corresponding stochastic cooperative game with players $[d]$ can be defined analogously to the deterministic case as $\nu_{p(f \mid \mathbf{D})} : \mathcal{X} \times 2^{[d]} \to \mathcal{L}_2(\mathbb{R})$ such that, for a given observation $\mathbf{x} \in \mathcal{X}$ and feature subset $S \subseteq [d]$, the stochastic payoff is

$$\nu_{p(f \mid \mathbf{D})}(\mathbf{x}, S) = \mathbb{E}_X [f(X) \mid X_S = \mathbf{x}_S]. \tag{3}$$

For brevity, we write $\nu_{p(f \mid \mathbf{D})}$ as $\nu_f$, bearing in mind that $\nu_f$ is a random function since $f$ is a random function. In particular, when $f$ is a GP with mean function $\tilde{m}$ and covariance function $\tilde{k}$, the stochastic payoff $\nu_f$ is also a GP indexed by $\mathbf{x}$ and $S$.

**Proposition 6** (Stochastic game $\nu_f$ as induced GP). *Let $f \sim \mathcal{GP}(\tilde{m}, \tilde{k})$ with integrable sample paths, i.e. $\int_{\mathcal{X}} |f| dp_X < \infty$ almost surely. The stochastic payoff function $\nu_f$ induced by $f$ is a Gaussian process with the following mean and covariance functions:*

$$m_\nu(\mathbf{x}, S) := \mathbb{E}_X[\tilde{m}(X) \mid X_S = \mathbf{x}_S], \tag{4}$$

$$k_\nu((\mathbf{x}, S), (\mathbf{x}', S')) := \mathbb{E}_{X, X'} \left[ \tilde{k}(X, X') \mid X_S = \mathbf{x}_S, X'_{S'} = \mathbf{x}'_{S'} \right]. \tag{5}$$

In other words, the stochastic game $\nu_f$ now assigns a payoff distribution to each feature subset $S$, where the variability arise from taking conditional expectation on a random function $f$. We note that this GP is also known as the conditional mean process, previously studied in Chau et al. [8, 18].

**Stochastic Shapley values as multivariate Gaussians for GPs.** Given the s-game $\nu_f$, we can characterise the corresponding SSVs using the weighted least squares formulation of DSVs [19, 2]. For each coalition $S_j \subseteq [d]$, let $z_j \in \{0,1\}^d$ be the binary vector such that $z_j[i] = 1$ if $i \in S_j$ and $\mathbf{Z} \in \{0,1\}^{2^d \times d}$ the concatenation of all $z_j$ vectors. When the context is clear, we use $\mathbf{Z}$ and $[d]$ interchangeably. Let $\mathbf{W} \in \mathbb{R}^{2^d \times 2^d}$ be the diagonal matrix with entries $W_{jj} = w(S_j) = \frac{d-1}{\binom{d}{|S_j|}|S_j|(d-|S_j|)}$ for all subsets with size $0 < |S_j| < d$. When $S_j$ has 0 or $d$ elements, the weights are set to $\infty$ to enforce the efficiency axiom.

**Theorem 7** (Stochastic Shapley values of $\nu_f$). *Let $\nu_f$ be an induced stochastic game from the GP $f \sim \mathcal{GP}(\tilde{m}, \tilde{k})$ and denote $\mathbf{v_x} := [\nu_f(\mathbf{x}, S_1), \ldots \nu_f(\mathbf{x}, S_{2^d})]^\top$ the vector of stochastic payoffs across all coalitions, then the corresponding stochastic Shapley values $\phi(\nu_f(\mathbf{x}, \cdot))$ follows a $d$-dimensional multivariate Gaussian distribution,*

$$\phi(\nu_f(\mathbf{x}, \cdot)) \sim \mathcal{N}(\mathbf{A}\mathbb{E}[\mathbf{v_x}], \mathbf{A}\mathbb{V}[v_x]\mathbf{A}^\top) \quad with \quad \mathbf{A} := (\mathbf{Z}^\top \mathbf{W}\mathbf{Z})^{-1}\mathbf{Z}^\top \mathbf{W}, \qquad (6)$$

*where $\mathbb{E}[\mathbf{v}_x] \in \mathbb{R}^{2^d}$ and $\mathbb{V}[\mathbf{v}_x] \in \mathbb{R}^{2^d \times 2^d}$ are the corresponding mean vector and covariance matrix of the payoffs.*

The derivation is straightforward using the fact that the stochastic payoffs $\mathbf{v_x}$ are multivariate Gaussian random variables and the matrix $\mathbf{A}$ is obtained from the weighted regression formulation of Shapley values. The resulting variance of $\phi(\nu_f(\mathbf{x}, \cdot))$ can then be interpreted as the uncertainties around the explanations, propagated from the posterior variance of $f$.

## 3.2 Estimation algorithms: the GP-SHAP and the BayesGP-SHAP

In this section, we introduce our main algorithm GP-SHAP and its variant BayesGP-SHAP to estimate the stochastic Shapley values for GP posterior distributions.

**Formulating GP-SHAP.** Given a set of observations $\mathbf{D} = \{(\mathbf{x}_i, y_i)\}_{i=1}^n = (\mathbf{X}, \mathbf{y})$ with GP prior $f \sim \mathcal{GP}(0, k)$, we can obtain the posterior GP $f \mid \mathbf{D} \sim \mathcal{GP}(\tilde{m}, \tilde{k})$ by using the Gaussian conditioning rule for regression or e.g. the variational or Laplace approximation for classification. In fact, the conditional expectations of mean and covariance function from a posterior GP can be estimated using conditional mean embeddings [38] without the need of explicit density estimations, following derivations of Chau et al. [8, 18]. We provide technical discussion on conditional mean embeddings in the appendix.

**Proposition 8** (Estimating $\nu_f$). *Given $\mathbf{D} = (\mathbf{X}, \mathbf{y})$ and the posterior GP $f \mid \mathbf{D} \sim \mathcal{GP}(\tilde{m}, \tilde{k})$, the mean and covariance function of the stochastic cooperative game $\nu_f$ can be estimated as,*

$$\hat{m}_\nu(\mathbf{x}, S) = \mathbf{b}(\mathbf{x}, S)^\top \tilde{m}(\mathbf{X}), \qquad \hat{k}_\nu\left((\mathbf{x}, S), (\mathbf{x}', S')\right) = \mathbf{b}(\mathbf{x}, S)^\top \tilde{\mathbf{K}}_{\mathbf{X}\mathbf{X}}\mathbf{b}(\mathbf{x}', S'), \qquad (7)$$

*where $\mathbf{b}(\mathbf{x}, S) := (\mathbf{K}_{\mathbf{X}_S\mathbf{X}_S} + \lambda I)^{-1}k_S(\mathbf{X}_S, \mathbf{x}_S)$, $\tilde{m}(\mathbf{X}) = [\tilde{m}(\mathbf{x}_1), \ldots, \tilde{m}(\mathbf{x}_n)]^\top$, and $k_S : \mathcal{X}_S \times \mathcal{X}_S \to \mathbb{R}$ is the kernel defined on the sub-feature space of $\mathcal{X}$ and we write $k_S(\mathbf{x}_S, \mathbf{X}_S) := [k_S(\mathbf{x}_S, \mathbf{x}_{1S}), ..., k_S(\mathbf{x}_S, \mathbf{x}_{nS})]$ and $\mathbf{K}_{\mathbf{X}\mathbf{X}}$ and $\tilde{\mathbf{K}}_{\mathbf{X}\mathbf{X}}$ as the gram matrix of $\mathbf{X}$ using kernel $k$ and $\tilde{k}$ respectively. The parameter $\lambda > 0$ is a fixed hyperparameter to stabilise the inversion.*

It is worth noting that the estimation $\hat{m}_\nu$ coincides with the one deployed in RKHS-SHAP [12] since we are also utilising conditional mean embeddings for the estimation of conditional expectation of RKHS functions. In addition, we obtain the analytical covariance function that is used to estimate the covariance of the stochastic Shapley values in the following proposition:

**Proposition 9** (GP-SHAP). *Let the matrix $\mathbf{A}$ be defined as in Theorem 7. The mean and covariance for the multivariate stochastic Shapley values can be estimated as,*

$$\phi(\hat{\nu}_f(\mathbf{x}, \cdot)) = \mathcal{N}\left(\mathbf{A}\mathbf{B}(\mathbf{x}, [d])^\top \tilde{m}(\mathbf{X}), \mathbf{A}\mathbf{B}(\mathbf{x}, [d])^\top \tilde{\mathbf{K}}_{\mathbf{X}\mathbf{X}}\mathbf{B}(\mathbf{x}, [d])\mathbf{A}^\top\right) \qquad (8)$$

*where $\mathbf{B}(\mathbf{x}, [d]) = [\mathbf{b}(\mathbf{x}, [d]_1), \ldots, \mathbf{b}(\mathbf{x}, [d]_{2^d})]^\top$.*

The complete algorithm, along with a discussion of computational techniques deployed to reduce computation time, such as vectorisation across $\mathbf{X}$ using tensor operations and incorporation of the sparse GP formulation to speed up computations of quantities from Propositions 8 and 9, is provided in the appendix. It is worth noting that subsampling coalitions to reduce computational cost while estimating Shapley values is also a standard approach in SHAP algorithm implementations [2]. Empirically, this procedure has been shown to give an unbiased estimate of the true DSVs, with the variance decreasing at a rate of $\mathcal{O}(\frac{1}{\ell})$, where $\ell$ is the number of coalition samples [22]. However, this approach incurs an additional source of uncertainty, due to estimation, and we propose to incorporate that as well into our framework by utilising the BayesSHAP approach proposed by Slack et al. [20].

**Formulating BayesGP-SHAP.** To capture the estimation uncertainty, Slack et al. [20] proposed BayesSHAP and reformulated the WLS procedure as a hierarchical Bayesian weighted least square and studied the corresponding posterior distribution over the deterministic Shapley values. We provide the hierarchical data generation process below, with an abuse of notations, **for a deterministic** $f$, we have

$$\bar{\nu} \mid z, \bar{\phi}, \epsilon, f, \mathbf{x} \sim \bar{\phi}^{\top} z + \epsilon \qquad \epsilon \mid z \sim \mathcal{N}(0, \sigma^2 w(z)^{-1}) \qquad (9)$$

$$\bar{\phi} \mid \sigma^2 \sim \mathcal{N}(0, \sigma^2 I) \qquad \sigma^2 \sim \text{Inv-}\chi^2(\ell_0, \sigma_0^2) \qquad (10)$$

where $w$ and $z$ are the weight function and binary vector respectively introduced in Theorem 7, and $\ell_0, \sigma_0^2$ are two hyperparameters, typically set to small values to keep priors uninformative. Slack et al. [20] showed that the posterior distribution on $\sigma^2$ and $\bar{\phi}$ follows a scaled Inv-$\chi^2$ and normal respectively, due to their corresponding conjugacies with the likelihood.

**Proposition 10** (BayesSHAP [20]). *Given the data generation above, the posterior distribution on $\bar{\phi}$ and $\sigma^2$ follows:*

$$\bar{\phi} \mid \sigma^2, \mathbf{Z}_\ell, f, \mathbf{x}, \mathbf{D} \sim \mathcal{N}(\mathbf{A}_\ell \bar{\mathbf{v}}_{\mathbf{x}}, (\mathbf{Z}_\ell^{\top} \mathbf{W}_\ell \mathbf{Z}_\ell)^{-1} \sigma^2) \qquad (11)$$

$$\sigma^2 \mid \mathbf{Z}_\ell, f, \mathbf{x}, \mathbf{D} \sim \text{Scaled-Inv-}\chi^2 \left( \ell_0 + \ell, \frac{\ell_0 \sigma_0^2 + \ell s^2(\bar{\mathbf{v}}_{\mathbf{x}})}{\ell_0 + \ell} \right) \qquad (12)$$

*where $\ell$ is the number of coalitions $\boldsymbol{\mathcal{S}} = \{S_j\}_{j=1}^{\ell}$ we sample uniformly from $2^{[d]}$, $\mathbf{Z}_\ell$ is the binary matrix representing $\boldsymbol{\mathcal{S}}$, and $\mathbf{W}_\ell$ is the corresponding weight matrix, and $\mathbf{A}_\ell = (\mathbf{Z}_\ell^{\top} \mathbf{W}_\ell \mathbf{Z}_\ell)^{-1} \mathbf{Z}_\ell^{\top} \mathbf{W}_\ell$ is the WLS matrix, $\bar{\mathbf{v}}_{\mathbf{x}} = [\bar{\nu}_f(\mathbf{x}, S_1), ..., \bar{\nu}_f(\mathbf{x}, S_\ell)]^{\top}$ is the vector of deterministic payoffs, and*

$$s^2(\bar{\mathbf{v}}_{\mathbf{x}}) = \frac{1}{\ell} \left[ (\bar{\mathbf{v}}_{\mathbf{x}} - \mathbf{Z}_\ell \mathbf{A}_\ell \bar{\mathbf{v}}_{\mathbf{x}})^{\top} W_\ell (\bar{\mathbf{v}}_{\mathbf{x}} - \mathbf{Z}_\ell \mathbf{A}_\ell \bar{\mathbf{v}}_{\mathbf{x}}) + (\mathbf{A}_\ell \bar{\mathbf{v}}_{\mathbf{x}})^{\top} (\mathbf{A}_\ell \bar{\mathbf{v}}_{\mathbf{x}}) \right] \qquad (13)$$

*measures the average weighted error in the regression and the norm of the mean explanations.*

It is important to highlight that while both GP-SHAP and BayesSHAP have a notion of the "posterior of Shapley values", the two sources of uncertainty which these approaches capture are very different: GP-SHAP corresponds to the predictive uncertainty induced by the GP posterior $p(f \mid \mathbf{D})$, whereas BayesSHAP is the uncertainty due to having to estimate Shapley values when their exact computation is infeasible due to the exponential number of coalitions – and is in Slack et al. [20] proposed for a deterministic $f$. Nonetheless, by integrating the BayesSHAP posterior of Shapley values through the posterior GP $p(f \mid \mathbf{D})$, we can in fact incorporate both sources of uncertainty, leading to our second algorithm, BayesGP-SHAP.

**Proposition 11** (**BayesGP-SHAP**). *Continuing from Propositions 9 and 10, the posterior distribution of the stochastic Shapley values can be estimated using the Bayesian WLS approach as,*

$$\phi \mid \sigma^2, \mathbf{Z}_\ell, \mathbf{x}, \mathbf{D} \sim \mathcal{N}\left( (\mathbf{A}_\ell \mathbf{B}(\mathbf{x}, \boldsymbol{\mathcal{S}}))^{\top} \tilde{m}(\mathbf{X}), \mathbf{A}_\ell \mathbf{B}(\mathbf{x}, \boldsymbol{\mathcal{S}})^{\top} \tilde{\mathbf{K}}_{\mathbf{X}\mathbf{X}} \mathbf{B}(\mathbf{x}, \boldsymbol{\mathcal{S}}) \mathbf{A}_\ell^{\top} + (\mathbf{Z}_\ell^{\top} \mathbf{W}_\ell \mathbf{Z}_\ell)^{-1} \sigma^2 \right)$$

*where $\sigma^2$ is sampled from $\sigma^2 \mid \mathbf{Z}_\ell \sim \text{Scaled-Inv-}\chi^2 \left( \ell_0 + \ell, \frac{\ell_0 \sigma_0^2 + \ell s^2(\mathbb{E}[\mathbf{v}_{\mathbf{x}}])}{\ell_0 + \ell} \right)$.*

We note that in the above proposition, instead of integrating $p(\sigma^2 \mid \mathbf{Z}_\ell, f, \mathbf{x}, \mathbf{D})$ with respect to the posterior GP, which leads to a complex scaled mixture of Gaussians, we simplify the expression and construct a scaled inverse chi-square distribution with $s^2(\mathbb{E}[\mathbf{v}_x])$ instead, which represents the error of the weighted regression with respect to the mean payoffs $\mathbb{E}[\mathbf{v}_x]$.

Conditionally on $\sigma^2$, the posterior variance in BayesGP-SHAP is therefore the sum of the variance from GP-SHAP and BayesSHAP due to Gaussian conjugacies.

## 4  Predictive explanations using the Shapley prior

In this section, we move beyond standard GP explanations by formulating explanation functions of a broader class of models as Gaussian processes. We introduce a *Shapley prior* over the space of vector-valued explanation functions $\phi : \mathcal{X} \to \mathbb{R}^d$, which correspond to Shapley values for arbitrary functions $f$. By treating previously obtained Shapley values as regression labels, we can predict explanations for new data without relying on the standard procedure to compute Shapley values. Our framework is not limited to explanations generated from GP-SHAP, but can also be applied to other explanation methods such as TreeSHAP, DeepSHAP, or model-agnostic KernelSHAP. The proposed approach learns the direct mapping from $\mathcal{X}$ to the space of Shapley values, without the need to access the underlying model $f$ during training, which is different from previous predictive approaches such as FastSHAP [39]. In contrast to prior work, Hill et al. [40] also considered fitting a GP regression model directly to explanation functions, but their focus is on incorporating the corresponding predictive uncertainty to explanations from black-box classifiers rather than predicting Shapley values for unseen data.

To achieve this, we leverage the key ideas behind the GP-SHAP algorithm, which uses the facts that GPs remain tractable under conditional expectations (to build the s-game) and linear combinations (to compute the stochastic Shapley values). By applying these properties to a GP prior $\mathcal{GP}(0, k)$ instead of a posterior, we have an induced prior over the space of Shapley functions $\phi$.

**Proposition 12** (The Shapley prior over $\phi$). *The prior $f \sim \mathcal{GP}(0, k)$ and the corresponding stochastic game $\nu_f(\mathbf{x}, S) = \mathbb{E}[f(X) \mid X_S = \mathbf{x}_S]$ induce a vector-valued GP prior over the explanation functions $\phi \sim \mathcal{GP}(0, \kappa)$ where $\kappa : \mathcal{X} \times \mathcal{X} \to \mathbb{R}^{d \times d}$ is the matrix-valued covariance kernel*

$$\kappa(\mathbf{x}, \mathbf{x}') = \mathcal{A}(\mathbf{x})^\top \mathcal{A}(\mathbf{x}'), \quad \mathcal{A}(\mathbf{x}) = \Psi(\mathbf{x}) \mathbf{A}^\top \tag{14}$$

*where $\Psi(\mathbf{x}) = \left[ \mathbb{E}[k(\cdot, X) \mid X_{S_1} = x_{S_1}], \dots, \mathbb{E}[k(\cdot, X) \mid X_{S_{2^d}} = x_{S_{2^d}}] \right]$, and the $^\top$ sign refers to taking inner products in the RKHS of $k$.*

By utilizing the Shapley function prior, we can apply our approach to predict explanations for a wide range of models, including trees, deep neural networks, or RKHS functions, by treating their explanations as noisy samples from this prior and using them as regression labels. Our approach is based on the perspective that there exists a true data generating mechanism $f_* : \mathcal{X} \to \mathcal{Y}$ and that any model $f$ we use is an approximation of $f_*$. Therefore, any explanations derived from $f$ can be seen as an attempt to reveal the true explanations under $f_*$. This perspective has also been adopted in the work of Chen et al. [36] and Marx et al. [14]. We present the predictive posterior below.

**Proposition 13** (Predictive explanations as multi-output GPs). *Given $\mathbf{D}_\phi = \{(\mathbf{x}_i, \phi_i)\}_{i=1}^n = (\mathbf{X}, \Phi_{\mathbf{X}})$ where $\phi_i \in \mathbb{R}^d$ are the Shapley values computed under predictive model $f$ and $\Phi_{\mathbf{X}} = [\phi_1, ..., \phi_n]^\top$, the predictive explanations for new data $\mathbf{x}'$ is distributed as,*

$$\phi(\mathbf{x}') \mid \mathbf{D}_\phi \sim \mathcal{N}\left( \tilde{m}_\phi(\mathbf{x}'), \quad \kappa(\mathbf{x}', \mathbf{x}') - \kappa(\mathbf{x}', \mathbf{X}) b_\kappa(\mathbf{x}', \mathbf{X}) \right) \tag{15}$$

*where $\tilde{m}_\phi(\mathbf{x}') = b_\kappa(\mathbf{x}', \mathbf{X})^\top \mathrm{vec}(\Phi_{\mathbf{X}})$, $b_\kappa(\mathbf{x}', \mathbf{X}) := (\mathcal{K}_{\mathbf{XX}} + \sigma_\phi^2 I)^{-1} \kappa(\mathbf{X}, \mathbf{x}')$, $\mathcal{K}_{\mathbf{XX}}$ is the gram matrix for kernel $\kappa$ of size $nd \times nd$, $\kappa(\mathbf{x}', \mathbf{X}) = [\kappa(\mathbf{x}', \mathbf{x}_1), \dots, \kappa(\mathbf{x}', \mathbf{x}_n)]$ is of size $d \times nd$ and $\sigma_\phi^2$ is the noise parameter for regression.*

In fact, we see that the posterior mean $\tilde{m}_\phi(\mathbf{x}')$ from (15) are Shapley values of the following payoff vector $\tilde{\mathbf{v}}_{\mathbf{x}'}$, computed based on the observed explanations $\Phi_{\mathbf{X}}$,

**Proposition 14** (Posterior mean as Shapley values for payoff vector $\tilde{\mathbf{v}}_{\mathbf{x}'}$). *The posterior mean $\tilde{m}_\phi(\mathbf{x}')$ corresponds to Shapley values for the payoff vector $\tilde{\mathbf{v}}_{\mathbf{x}'}$, i.e., $\tilde{m}_\phi(\mathbf{x}') = \mathbf{A} \tilde{\mathbf{v}}_{\mathbf{x}'}$, where $\tilde{\mathbf{v}}_{\mathbf{x}'} = \sum_{i=1}^n \Psi(\mathbf{x}')^\top \Psi(\mathbf{x}_i) \mathbf{A}^\top \alpha_i$ and $\alpha_i \in \mathbb{R}^d$ is the $[i, ..., i + (d-1)]$ subvector of $(\mathcal{K}_{\mathbf{XX}} + \sigma_\phi^2 I)^{-1} \mathrm{vec}(\Phi_{\mathbf{X}})$.*

## 5  Illustrations

We demonstrate the proposed approaches through three sets of illustrations. We first conduct an ablation study to examine how predictive and estimation uncertainty captured by our explanation algorithms vary under different configurations. We then move on to discuss various exploratory tools that can assist practitioners in utilizing the stochastic explanations. Finally, we demonstrate the

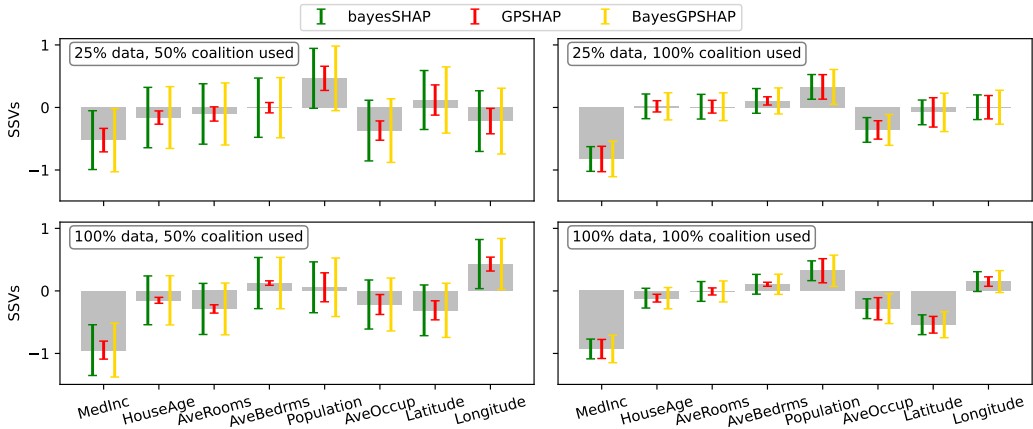

Figure 1: Ablation study on different uncertainties captured by GP-SHAP, BayesSHAP, and BayesGP-SHAP when computing local explanations (SSVs) using the California housing dataset [41]. 95% credible intervals around explanations are shown.

effectiveness of the Shapley prior for the predictive explanations problem. Our code is included in the supplementary material, and we provide implementation details in the appendix. For all experiments, we pick the radial basis function (RBF) as our covariance kernel $k$ for the GPs.

## 5.1 Ablation study on different notions of uncertainties captured

We conducted a comparison between GP-SHAP, BayesSHAP, and BayesGP-SHAP to demonstrate the differences between model predictive uncertainty and estimation uncertainty captured in the stochastic Shapley values. BayesSHAP is applied to the predictive posterior mean of GP to obtain its explanation. For this purpose, we used the California housing dataset [41] from the StatlLib repository, which includes $20640$ instances and $8$ numerical features, with the goal of predicting the median house value for California districts, expressed in hundreds of thousands of dollars. We trained our GP model using $25\%$ and $100\%$ of the data and calculated local stochastic explanations from GP-SHAP, BayesSHAP, and BayesGP-SHAP using $50\%$ and $100\%$ of coalitions. The results, shown in Figure 1, demonstrate that the magnitude of BayesSHAP uncertainties (green bars) are uniform across features as it is designed to capture the overall estimation performance and not feature-specific uncertainty. In contrast, GP-SHAP (red bars) exhibits varying uncertainties across features due to the propagation of variation from the posterior GP $f$ to the attributions. This allows practitioners to make more granular statements about the uncertainty around specific feature explanations. We also observe that increasing the number of coalitions and training data generally leads to a decrease in both estimation and predictive uncertainties, but the estimation uncertainty drops more significantly. BayesGP-SHAP (yellow bars) consistently provides a more conservative uncertainty estimation by considering both predictive and estimation uncertainties.

## 5.2 Exploratory analysis of the stochastic explanations

We propose several exploratory analysis methods to aid practitioners in comprehending the stochastic explanations generated for their downstream tasks. To this end, we use BayesGP-SHAP to explain a Gaussian process model trained on the breast cancer dataset [42]. The dataset consists of $569$ patients, $30$ numeric features, and the objective is to predict whether a tumor is malignant or benign.

**Local explanation.** To visualize local explanations, we can plot the mean and standard deviations of stochastic Shapley values in a bar plot. This approach allows us to not only understand the degree of contribution a feature has to the prediction but also the corresponding credible interval as a measure of explanation uncertainty. Figure 2a displays that both "worst fractal dimension" and "worst perimeter" features have a similar contribution in terms of their absolute mean stochastic Shapley values. However, the model is much more uncertain about the former, allowing the user to calibrate their trust in model explanations.

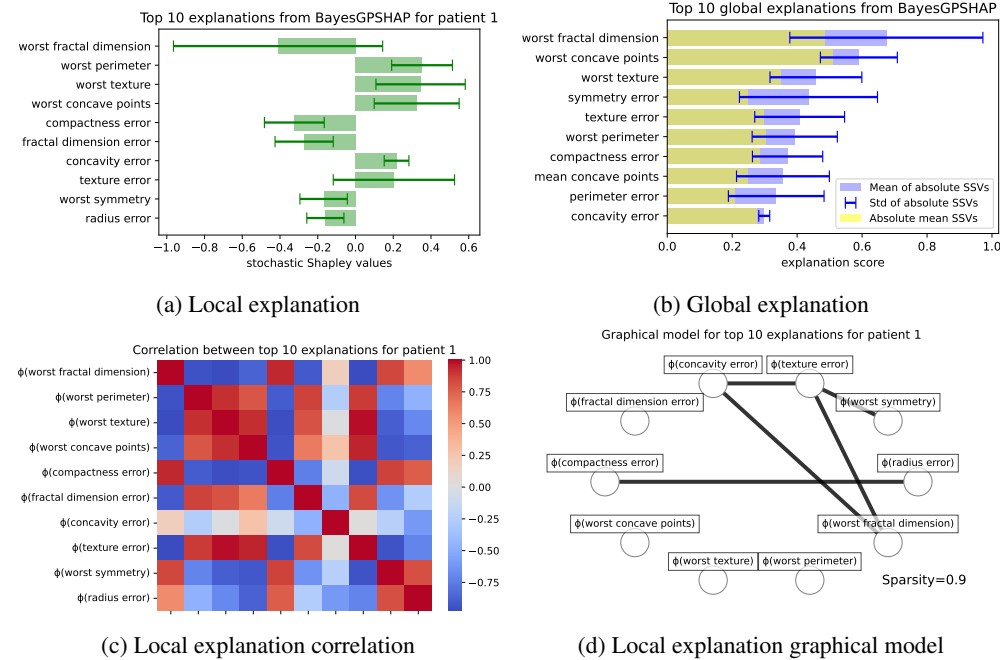

|  |  |
|---|---|
| (a) Local explanation | (b) Global explanation |
| (c) Local explanation correlation | (d) Local explanation graphical model |

Figure 2: Illustrations of possible exploratory analyses utilising the BayesGP-SHAP covariance structure, applied on the breast cancer dataset.

**Heuristic global explanation as means of absolute stochastic Shapley values.** It is common to compute the mean absolute DSVs as a proxy to global feature contributions [43] based on computed local explanations. Although this heuristic does not result in any explanations that are themselves DSVs of any game at the global level, they provide a quick summary to practitioners about the overall contributions. We can similarly compute the average of the absolute stochastic Shapley values, which are now distributed according to a folded multivariate Gaussian distribution [44] with a tractable covariance structure. However, it is important to highlight a key distinction from the previous approach: when computing the **mean of absolute SSVs**, we consider the uncertainty in the local stochastic explanations, while directly calculating the **absolute values of mean SSVs** (equivalent to computing absolute DSVs) disregards this uncertainty. We believe the former approach is more appropriate than the latter as practitioners utilising Gaussian processes would want their global explanations to take the predictive uncertainty into account. The blue and yellow bars in Figure 2b depict these two approaches, respectively. Notably, we observe that the blue bars suggest that the "worst fractal dimension" feature is considered more influential than "worst concave points" because it accounts for the higher variability in the former feature's explanation. In contrast, the yellow bar overlooks this variability, leading to a different conclusion.

**Correlations and dependencies across explanations.** As we have access to the explanation covariance, we can explore their correlation and visualise them as in Figure 2c. For instance, the stochastic Shapley values of "concavity error" are less correlated with other features. Moreover, as our explanations are multivariate Gaussians, we can build an undirected Gaussian graphical model [45] using the precision matrix (inverse of the covariance matrix) to study the independence across the local stochastic explanations. We visualise the corresponding graphical model for explanations from patient 1 in Figure 2d by setting a sparsity threshold of $90\%$, following the approach in [46].

## 5.3 Predictive explanations

Finally, we showcase the effectiveness of our Shapley prior by comparing the regression performances of a multi-output GP using the Shapley prior with multi-output random forest and neural networks when predicting withheld explanations generated from GP-SHAP, Tree-SHAP, and DeepSHAP, respectively. We use the diabetes dataset [47] from the UCI repository, which contains $442$ patients with $10$ numerical features and the goal is to predict disease progression. We first train a GP, a random forest, and a neural network, to obtain the

subsequent explanations from GP-SHAP, TreeSHAP, and DeepSHAP. Next, we feed $70\%$ of the explanations to our predictive model as training data and the remaining $30\%$ as test data. We repeat this process over $10$ seeds and compute the root mean squared error between the predicted explanation and the exact explanations for respective models, as shown in Figure 3. We observe that our GP-model with the Shapley prior consistently outperforms random forest and neural network model for predicting all three types of explanations. This demonstrates that the inductive bias that comes from our choice of covariance structure $\kappa$ allows to build more accurate predictive explanation models. We also observe that the overall average prediction error for explanations generated from GP-SHAP is lower than that of TreeSHAP [11] and DeepSHAP [2], suggesting that GP-SHAP produce explanations that are easier to learn. This follows our intuition as GPs are typically smoother functions than trees and neural networks.

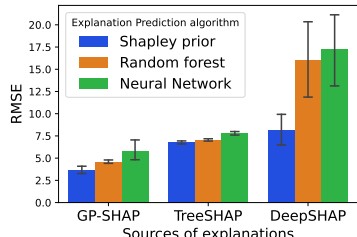

Figure 3: Predictive performance of using Shapley prior to predict explanations generated from different explanation algorithms on the diabetes dataset.

## 6    Discussion

In this work, we presented a novel and principled approach for explaining Gaussian process models using stochastic Shapley values. The proposed algorithm GP-SHAP, and its variant BayesGP-SHAP, allow practitioners to consider both predictive and estimation uncertainties when reasoning about the explanations and calibrate their trust in the model accordingly. Furthermore, we consider the setting of predictive explanations, where we introduced a Shapley prior over explanation functions, enabling us to model and predict Shapley values for a wider range of predictive models. We demonstrated the effectiveness of our methods through a number of illustrations and discussed various exploratory tools for practitioners to analyse the stochastic explanations.

**Limitations and future outlook.**  While the general framework of using stochastic Shapley values for stochastic explanations can be applied beyond Gaussian process models, the specific estimation algorithms presented in this paper are tailored to GPs and may not be directly applicable to other probabilistic models like Bayesian neural networks. These alternative models would require different estimation procedures as the resulting explanations would not be GPs anymore. Another promising avenue for future research involves leveraging the uncertainty obtained from predicting explanations and incorporating it into Bayesian optimization for guiding experimental design. For example, this could allow practitioners to explore data regions that produce significant Shapley values for specific features. At last, we would like to highlight that although SHAP is popular, as an explanation tool it contains several limitations, specifically Kumar et al. [48] have shown that SHAP violates several expected properties of explanations from a human-centric perspective.

## Acknowledgement

The authors would like to thank Jean Francois Ton, Shahine Bouabid, Jake Fawkes, and Simon Föll for insightful discussions. The authors are also indebted to anonymous reviewers whose feedback has significantly improved the manuscript.

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

SUPPLEMENTARY MATERIAL: **Explaining the uncertain: Stochastic Shapley values for Gaussian process models**

## A   The GP-SHAP algorithm and discussion on computation techniques

We present the complete algorithm for both GP-SHAP and BayesGP-SHAP in Algorithm 1.

---

**Algorithm 1** GP-SHAP / BayesGP-SHAP

---

**Input:** Posterior mean function $\tilde{m}$, posterior covariance function $\tilde{k}$, inducing locations $\tilde{\mathbf{X}}$, explanation instances $\mathbf{X}$, number of coalition samples $n_Z$, hyperparameter $\lambda, n_0, \sigma_0^2$, base kernel $k$, algorithm **algo**,

1: Compute $n_I$ = number of inducing location, $n$ = number of explanation instances, $d$ = number of features.
2: Compute Cholesky decomposition on posterior covariance $\mathbf{L}\mathbf{L}^\top = \tilde{\mathbf{K}}_{\tilde{\mathbf{X}}\tilde{\mathbf{X}}}$
3: Sample coalitions $\mathcal{S} = \{S_1, ..., S_{n_Z}\}$ from $[d]$, build binary matrix $\mathbf{Z} = \{0,1\}^{n_Z \times d}$ from $\mathcal{S}$, and compute weights $W = \mathrm{diag}[w_1, ..., w_{n_Z}]$ with $w_i = \frac{d-1}{\binom{d}{|S_i|}|S_i|(d-|S_i|)}$.
4: Compute $\mathbf{A} = (\mathbf{Z}^\top W \mathbf{Z})^{-1} \mathbf{Z}^\top W$  ▷ Shape: $d \times n_Z$
5: Compute $\mathbf{B}(\mathbf{X}, \mathcal{S}) = [(\mathbf{K}_{\tilde{\mathbf{X}}_S \tilde{\mathbf{X}}_S} + \lambda I)^{-1} k_S(\tilde{\mathbf{X}}_S, \mathbf{X}_S) \quad \textbf{for } S \text{ in } \mathcal{S}]$  ▷ Shape: $n_Z \times n_I \times n$
6: Compute $\mathbf{Q}$ where $\mathbf{Q}_{i,l,k} = \sum_j \mathbf{B}(\mathbf{X}, \mathcal{S})_{i,j,k} \mathbf{L}_{j,l}$  ▷ Shape: $n_Z \times n \times n_I$
7: Compute $\mathbf{R}$ where $\mathbf{R}_{i,k,l} = \sum_j \mathbf{A}_{i,j} \mathbf{Q}_{j,k,l}$  ▷ Shape: $d \times n \times n_I$
8: Compute $\mathbf{V}$ where $\mathbf{V}_{i,m,k,n} = \sum_{j,l} \mathbf{R}_{i,j,k} \mathbf{R}_{m,l,n}$  ▷ Shape: $d \times d \times n \times n$
9: Compute $\mathbf{E}$ where $\mathbf{E}_{i,k} = \sum_j \mathbf{B}(\mathbf{X}, \mathcal{S})_{i,j,k} \tilde{m}(\tilde{\mathbf{X}})_j$  ▷ Shape: $n_Z \times n$
10: Compute $\Phi = \mathbf{A}\mathbf{E}$  ▷ The mean stochastic Shapley values of shape $d \times n$
11: **if algo** = GP-SHAP **then**
12:     **return** mean explanations $\Phi$ and covariance $\mathbf{V}$ between $d$ features and $n$ instances
13: **else if algo** = BayesGP-SHAP **then**
14:     Compute $s^2 = \mathrm{diag}\left((\mathbf{E} - \mathbf{Z}\Phi)^\top \mathbf{W} (\mathbf{E} - \mathbf{Z}\Phi)\right) + \mathrm{diag}(\Phi^\top \Phi)$  ▷ Shape: $n \times 1$
15:     Sample $\sigma^2$ from Scaled-Inv-$\chi^2 \left(n_0 + n_Z, \frac{n_0 \sigma_0^2 + n_Z s^2}{n_0 + n_Z}\right)$  ▷ Shape: $n \times 1$
16:     **return** mean explanations $\Phi$ and covariance $\mathbf{V} + (\mathbf{Z}^\top \mathbf{W} \mathbf{Z})^{-1} \sigma^2$
17: **end if**

---

**Computational considerations.**   In terms of computational complexity, one of the most demanding operations in the algorithm is the computation of conditional mean embeddings in step 5. Instead of naively inverting an $n \times n$ matrix, which would have a computational cost of $\mathcal{O}(n^3)$, we employ the conjugate gradient method to reduce the computation of the conditional mean embedding component to $\mathcal{O}(n^2 a)$, where $a \ll n$ represents the number of conjugate gradient iterations. Additionally, to further reduce runtime, we utilize the variational sparse GP model [49]. This model learns a set of inducing locations $\tilde{\mathbf{X}}$ with a size of $n_I \ll n$, which can be reused for the estimation of conditional mean embeddings in the algorithm. Consequently, the computation of the conditional expectation is reduced from $\mathcal{O}(n^2 a)$ to $\mathcal{O}(n_I^2 a)$. Another computational burden arises from the computation of the full covariance matrix across $d$ features and $n$ instances, which requires storage of a $n^2 d^2$ matrix. However, since the full covariance matrix can be factorized into the $\mathbf{R}$ component from step 7 of the algorithm, we can store this low-rank component and compute covariances between specific instances when necessary. It is worth noting that this decomposition of the covariance matrix allows us to avoid redundant computations when computing the covariance component, as we no longer need to iterate over all possible coalitions twice. Finally, we can further speed up our computational by parallelising computation across the sub-sampled coalitions in step 5.

# B   Proofs and derivations

## B.1   Section 2 proofs: Stochastic Shapley values

We include the full proof of the derivation of stochastic Shapley values for completeness. The proof is analogous to the original work of Shapley's [1] but extended to random variable payoffs. Ma et al. [16] has also proved the same theorem but used a different proving strategy. They started with the solution and showed it satisfies the axioms and then prove uniqueness, whereas the following proof starts from the characterisation of s-games and derive the solution from a bottom-up fashion.

To facilitate the proof, we first introduce the concept of stochastic symmetric game.

**Proposition 15** (s-symmetric games). *Let $C$ be a real-valued random variables, then the symmetric game $\nu_{C,R}(S) := C\mathbf{1}[R \subseteq S]$ gets a stochastic shapley value as,*

$$\phi_i(\nu_{C,R}) = \frac{C}{r} \tag{16}$$

*where $r = |R|$.*

*Proof.* Take any $i, j \in R$, pick a permutation $\pi \in \Pi(U)$ so that $\pi R = R$ and $\pi i = j$, so the induced game $\pi \nu_{C,R} = \nu_{C,R}$, and therefore by the s-symmetry axiom,

$$\phi_j(\nu_{C,R}) = \phi_i(\nu_{C,R}) \tag{17}$$

Now by the s-efficiency axiom,

$$C = \nu_{C,R}(R) = \sum_{j \in R} \phi_j(\nu_{C,R}) = r\phi_i(\nu_{C,R}) \tag{18}$$

for any $i \in R$.   □

Now we can characterise the form of any stochastic game as follows:

**Proposition 16.** *All s-games with finite carrier can be written as a linear combination of s-symmetric games,*

$$\nu = \sum_{R \subseteq N, R \neq \emptyset} \nu_{c_R(\nu),R} \tag{19}$$

*where*

$$C_R(\nu) = \sum_{T \subseteq R} (-1)^{r-t} \nu(T) \tag{20}$$

*Proof.* We start by verifying

$$\nu(S) = \sum_{R \subseteq N, R \neq \emptyset} \nu_{c_R(\nu),R}(S) \tag{21}$$

holds for all $S \subseteq U$, and for any finite carrier $N$ of $\nu$. If $S \subseteq N$, then we can rewrite the expression as,

$$\nu(S) = \sum_{R \subseteq S} \sum_{T \subseteq R} (-1)^{r-t} \nu(T) \tag{22}$$

$$= \sum_{T \subseteq S} \sum_{T \subseteq R \subseteq S} (-1)^{r-t} \nu(T) \tag{23}$$

$$= \sum_{T \subseteq S} \nu(T) \sum_{r=t}^{s} (-1)^{r-t} \binom{s-t}{r-t} \tag{24}$$

$$= \nu(S) \tag{25}$$

where in the last equation we used the fact that $\sum_{r=t}^{s} (-1)^{r-t} \binom{s-t}{r-t}$ is a binomal expansion of $(1 + (-1))^{s-t}$, therefore the only non-zero expression is when $t = s$.

□

We can now prove the uniqueness of stochastic Shapley values,

**Theorem 4** (Stochastic Shapley values)**.** *The only stochastic value allocation $\phi$ of $\nu$ satisfying s-symmetry, s-efficiency, and s-linearity takes the following form,*

$$\phi_i(\nu) = \sum_{S \subseteq N \setminus \{i\}} c_{|S|} \left( \nu(S \cup i) - \nu(S) \right) \tag{1}$$

*where $N$ is the smallest carrier set of $\Omega$, $c_{|S|} = \frac{1}{|N|} \binom{|N|-1}{|S|}^{-1}$ and $\phi_i(\nu)$ is the $i^{th}$ SSV of s-game $\nu$.*

*Proof.* First, let us denote

$$\gamma_i(S) := \sum_{\substack{R \subseteq N \\ S \cup \{i\} \subseteq R}} (-1)^{r-s} \frac{1}{r}.$$

Applying the s-linearity axiom on $\phi$ to the characterisation of $\nu$ from the previous propositions leads us to the following,

$$\phi_i(\nu) = \phi_i \left( \sum_{R \subseteq N, R \neq \emptyset} \nu_{C_R(\nu), R} \right) \tag{26}$$

$$= \sum_{R \subseteq N, R \neq \emptyset} \phi_i(\nu_{C_R(\nu), R}) \tag{27}$$

$$= \sum_{R \subseteq N, i \in R} c_R(\nu) \frac{1}{r} \tag{28}$$

$$= \sum_{R \subseteq N, i \in R} \frac{1}{r} \left( \sum_{S \subseteq R} (-1)^{r-s} \nu(S) \right) \tag{29}$$

$$= \sum_{S \subseteq N} \sum_{\substack{R \subseteq N \\ S \cup \{i\} \subseteq R}} (-1)^{r-s} \nu(S) \frac{1}{r} \tag{30}$$

$$= \sum_{S \subseteq N} \gamma_i(S) \nu(S) \tag{31}$$

$$= \sum_{\substack{S \subseteq N \\ i \in S}} \gamma_i(S) \nu(S) + \gamma_i(S - \{i\}) \nu(S - \{i\}) \tag{32}$$

$$= \sum_{\substack{S \subseteq N \\ i \in S}} \gamma_i(S) \left( \nu(S) - \nu(S - \{i\}) \right) \tag{33}$$

$$= \sum_{\substack{S \subseteq N \\ i \in S}} \frac{(s-1)!(n-s)!}{n!} \left( \nu(S) - \nu(S - \{i\}) \right) \tag{34}$$

$$= \sum_{S \subseteq N \setminus \{i\}} c_{|S|} \left( \nu(S \cup i) - \nu(S) \right) \tag{35}$$

where in (32) we used the following observation: given $i \notin S' \subseteq N$, and $S = S' \cup \{i\}$, then $\gamma_i(S) = -\gamma_i(S')$.

It satisfies uniqueness by construction. $\qquad \square$

**Proposition 5.** *Given the player set $\Omega$, let $\nu$ be a stochastic game, $\phi$ a stochastic Shapley value allocation, and $\bar{\phi}$ a deterministic Shapley value allocation. Suppose that $\mathbb{E}[\nu]$ and $\mathbb{V}[\nu]$ are the corresponding mean and variance d-games, respectively. Then, $\mathbb{E}[\phi(\nu)] = \bar{\phi}(\mathbb{E}[\nu])$, but $\mathbb{V}[\phi(\nu)] \neq \bar{\phi}(\mathbb{V}[\nu])$. In particular, the SSV variance is given by*

$$\mathbb{V}[\phi_i(\nu)] = \sum_{S \subseteq N \setminus \{i\}} \sum_{S' \subseteq N \setminus \{i\}} c_{|S|} c_{|S'|} \left( \mathbb{C}[\nu_{S \cup i}, \nu_{S' \cup i}] - \mathbb{C}[\nu_{S \cup i}, \nu_{S'}] - \mathbb{C}[\nu_S, \nu_{S' \cup i}] + \mathbb{C}[\nu_S, \nu_{S'}] \right),$$

*where $\nu_S = \nu(S)$ and $\mathbb{C}$ is the covariance function between the stochastic payoffs.*

*Proof.* The equivalence between mean of stochastic Shapley values and deterministic Shapley values of mean game is trivial to show leveraging the linearity of expectation. The variance of $\mathbb{V}[\phi_i(\nu)]$ can be shown by repeatedly applying the standard identity $\mathbb{V}[X+Y] = \mathbb{V}[X] + \mathbb{V}[Y] + 2\mathbb{C}[X,Y]$ for random variables $X, Y$. Now consider the deterministic Shapley values of variance game $\mathbb{V}[\nu]$,

$$\bar{\phi}_i[\mathbb{V}[\nu(\cdot)]] = \sum_{S \subseteq N \setminus \{i\}} c_{|S|} \left( \mathbb{V}[\nu(S \cup i)] - \mathbb{V}[\nu(S)] \right) \tag{36}$$

Comparing to the expression of $\mathbb{V}[\phi_i(\nu)]$ from the lemma,

$$\mathbb{V}[\phi_i(\nu)] = \sum_{S \subseteq N \setminus \{i\}} \sum_{S' \subseteq N \setminus \{i\}} c_{|S|} c_{|S'|} \big( \mathbb{C}[\nu_{S \cup i}, \nu_{S' \cup i}] - \mathbb{C}[\nu_{S \cup i}, \nu_{S'}] - \mathbb{C}[\nu_S, \nu_{S' \cup i}] + \mathbb{C}[\nu_S, \nu_{S'}] \big),$$

even if we assume mutual independence across all payoff random variables, leading to $\mathbb{C}[\nu(S \cup i), \nu(S)] = 0$ for all $S$, we still would not subtract but instead sum the variance of $\mathbb{V}[\nu(S \cup i)]$ and $\mathbb{V}[\nu(S)]$. Therefore the variances of stochastic Shapley values is not the same as the deterministic Shapley values of the variance game. $\square$

### B.2 Section 3.1 proofs on the stochastic Shapley values for induced stochastic game from GP

**Proposition 6** (Stochastic game $\nu_f$ as induced GP). *Let $f \sim \mathcal{GP}(\tilde{m}, \tilde{k})$ with integrable sample paths, i.e. $\int_{\mathcal{X}} |f| dp_X < \infty$ almost surely. The stochastic payoff function $\nu_f$ induced by $f$ is a Gaussian process with the following mean and covariance functions:*

$$m_\nu(\mathbf{x}, S) := \mathbb{E}_X[\tilde{m}(X) \mid X_S = \mathbf{x}_S], \tag{4}$$

$$k_\nu \left( (\mathbf{x}, S), (\mathbf{x}', S') \right) := \mathbb{E}_{X, X'} \left[ \tilde{k}(X, X') \mid X_S = \mathbf{x}_S, X'_{S'} = \mathbf{x}'_{S'} \right]. \tag{5}$$

*Proof.* This is a direct application of Chau et al. [18, Proposition 3.2] to the distribution $P(X \mid X_S = \mathbf{x}_S)$. $\square$

**Theorem 7** (Stochastic Shapley values of $\nu_f$). *Let $\nu_f$ be an induced stochastic game from the GP $f \sim \mathcal{GP}(\tilde{m}, \tilde{k})$ and denote $\mathbf{v}_\mathbf{x} := [\nu_f(\mathbf{x}, S_1), \ldots \nu_f(\mathbf{x}, S_{2^d})]^\top$ the vector of stochastic payoffs across all coalitions, then the corresponding stochastic Shapley values $\phi(\nu_f(\mathbf{x}, \cdot))$ follows a $d$-dimensional multivariate Gaussian distribution,*

$$\phi(\nu_f(\mathbf{x}, \cdot)) \sim \mathcal{N}(\mathbf{A}\mathbb{E}[\mathbf{v}_\mathbf{x}], \mathbf{A}\mathbb{V}[\mathbf{v}_x]\mathbf{A}^\top) \quad \text{with} \quad \mathbf{A} := (\mathbf{Z}^\top \mathbf{W} \mathbf{Z})^{-1} \mathbf{Z}^\top \mathbf{W}, \tag{6}$$

*where $\mathbb{E}[\mathbf{v}_x] \in \mathbb{R}^{2^d}$ and $\mathbb{V}[\mathbf{v}_x] \in \mathbb{R}^{2^d \times 2^d}$ are the corresponding mean vector and covariance matrix of the payoffs.*

*Proof.* Recall from Lundberg and Lee [2, Theorem 2], for deterministic Shapley values, given a deterministic payoff $\bar{\mathbf{v}}_\mathbf{x}$ for all $2^d$ coalitions, the expression of Shapley values for each $i \in [d]$,

$$\bar{\phi}_{\mathbf{x}i} = \sum_{S \subseteq [d] \setminus \{i\}} c_{|S|} \left( \bar{\nu}_f(S \cup i) - \bar{\nu}_f(S) \right) \tag{37}$$

can be written compactly as the following vector,

$$\bar{\phi}_\mathbf{x} = \mathbf{A}\bar{\mathbf{v}}_\mathbf{x}. \tag{38}$$

We can therefore similarly write down the form of the stochastic Shapley values using this linear operator $\mathbf{A}$, acting now on a vector of random variable output stochastic payoff vector $\mathbf{v}_\mathbf{x}$,

$$\phi_\mathbf{x} = \mathbf{A}\mathbf{v}_\mathbf{x}. \tag{39}$$

Nonetheless, as Proposition 8 implies that $\mathbf{v}_\mathbf{x}$ is a multivariate Gaussian, therefore $\phi_\mathbf{x}$ is also multivariate Gaussian with mean and covariance the following,

$$\mathbf{v}_\mathbf{x} \sim \mathcal{N} \left( A\mathbb{E}[\mathbf{v}_\mathbf{x}], A\mathbb{V}[\mathbf{v}_\mathbf{x}]A^\top \right). \tag{40}$$

$\square$

## B.3 Section 3.2 proofs on estimation

To proceed, we first introduce the concepts of conditional mean embedding as a tool to estimate conditional expectation of functions living in their corresponding RKHSs,

**Definition 17** (Conditional mean embedding [38]). *Let $X, Y$ be random variables and $k : \mathcal{X} \to \mathcal{X} \to \mathbb{R}$ a kernel on $X$, then we define the following as the conditional mean embedding of $p(X \mid Y = y)$,*

$$\mu_{X|Y=y} := \int k(\cdot, X) d\mathbb{P}(X \mid Y = y) \tag{41}$$

**Proposition 18** (Conditional Mean estimation). *For random variable $X, Y$, and a kernel $k : \mathcal{X} \to \mathcal{X} \to \mathbb{R}$ on $\mathcal{X}$ and a kernel $l : \mathcal{Y} \to \mathcal{Y} \to \mathbb{R}$ on $\mathcal{Y}$. Given observations $\mathbf{D} = \{\mathbf{X}, \mathbf{y}\}$, the empirical conditional mean embedding can be estimated as*

$$\hat{\mu}_{X|Y=y} = l(y, \mathbf{y}) \left(\mathbf{L_{yy}} + \lambda I\right)^{-1} k(\mathbf{X}, \cdot), \tag{42}$$

*where $l(\cdot, \mathbf{y}) = [l(y, y_1), \ldots, l(y, y_n)]^\top$ and $k(\cdot, \mathbf{X}) = [k(\cdot, \mathbf{x}_1), \ldots, k(\cdot, \mathbf{x}_n)]^\top$, the parameter $\lambda > 0$ is there to stablise the inversion. Now for $f \in \mathcal{H}_k$, the conditional expectation can then be estimated as,*

$$\hat{\mathbb{E}}[f(X) \mid Y = y] = \langle \hat{\mu}_{X|Y=y}, f \rangle \tag{43}$$

$$= l(y, \mathbf{y})(\mathbf{L_{yy}} + \lambda I)^{-1}\mathbf{f}, \tag{44}$$

*where $\mathbf{f} = [f(\mathbf{x}_1), \ldots, f(\mathbf{x}_n)]^\top$.*

*Proof.* This is standard result from literature, please read Song et al. [50], Muandet et al. [38] for more details. $\square$

Now we can apply these propositions to estimate the mean and covariance functions of the induced stochastic game from GP,

**Proposition 8** (Estimating $\nu_f$). *Given $\mathbf{D} = (\mathbf{X}, \mathbf{y})$ and the posterior GP $f \mid \mathbf{D} \sim \mathcal{GP}(\tilde{m}, \tilde{k})$, the mean and covariance function of the stochastic cooperative game $\nu_f$ can be estimated as,*

$$\hat{m}_\nu(\mathbf{x}, S) = \mathbf{b}(\mathbf{x}, S)^\top \tilde{m}(\mathbf{X}), \qquad \hat{k}_\nu\left((\mathbf{x}, S), (\mathbf{x}', S')\right) = \mathbf{b}(\mathbf{x}, S)^\top \tilde{\mathbf{K}}_{\mathbf{XX}} \mathbf{b}(\mathbf{x}', S'), \tag{7}$$

*where $\mathbf{b}(\mathbf{x}, S) := (\mathbf{K}_{\mathbf{X}_S\mathbf{X}_S} + \lambda I)^{-1} k_S(\mathbf{X}_S, \mathbf{x}_S)$, $\tilde{m}(\mathbf{X}) = [\tilde{m}(\mathbf{x}_1), \ldots, \tilde{m}(\mathbf{x}_n)]^\top$, and $k_S : \mathcal{X}_S \times \mathcal{X}_S \to \mathbb{R}$ is the kernel defined on the sub-feature space of $\mathcal{X}$ and we write $k_S(\mathbf{x}_S, \mathbf{X}_S) := [k_S(\mathbf{x}_S, \mathbf{x}_{1S}), ..., k_S(\mathbf{x}_S, \mathbf{x}_{nS})]$ and $\mathbf{K}_{\mathbf{XX}}$ and $\tilde{\mathbf{K}}_{\mathbf{XX}}$ as the gram matrix of $\mathbf{X}$ using kernel $k$ and $\tilde{k}$ respectively. The parameter $\lambda > 0$ is a fixed hyperparameter to stabilise the inversion.*

*Proof.* Without loss of generality, we will demonstrate this proposition with $\tilde{m}, \tilde{k}$ obtained via standard GP regression, i.e.,

$$\tilde{m}(\mathbf{x}) = k(\mathbf{x}, \mathbf{X})(\mathbf{K}_{\mathbf{XX}} + \sigma^2 I)^{-1}\mathbf{y} \tag{45}$$

$$\tilde{k}(\mathbf{x}, \mathbf{x}') = k(\mathbf{x}, \mathbf{x}') - k(\mathbf{x}, \mathbf{X})(\mathbf{K}_{\mathbf{XX}} + \sigma^2)^{-1} k(\mathbf{X}, \mathbf{x}'). \tag{46}$$

Starting with the mean function,

$$\mathbb{E}[\tilde{m}(X) \mid X_S = \mathbf{x}_S] = \mathbb{E}_X[k(X, \mathbf{X})(\mathbf{K}_{\mathbf{XX}} + \sigma^2 I)^{-1}\mathbf{y} \mid X_S = \mathbf{x}_S] \tag{47}$$

$$= \langle k(\cdot, \mathbf{X})(\mathbf{K}_{\mathbf{XX}} + \sigma^2 I)^{-1}\mathbf{y}, \mu_{X|X_S=\mathbf{x}_S}\rangle_{\mathcal{H}_k}. \tag{48}$$

We can replace the population conditional mean embedding with the empirical version, and expand,

$$\hat{\mathbb{E}}[\tilde{m}(X) \mid X_S = \mathbf{x}_S] = \langle k(\cdot, \mathbf{X})(\mathbf{K}_{\mathbf{XX}} + \sigma^2 I)^{-1}\mathbf{y}, \hat{\mu}_{X|X_S=\mathbf{x}_S}\rangle_{\mathcal{H}_k} \tag{49}$$

$$= k_S(\mathbf{X}_S, \mathbf{x}_S)(\mathbf{K}_{\mathbf{X}_S\mathbf{X}_S} + \lambda I)^{-1}\mathbf{K}_{\mathbf{XX}}(\mathbf{K}_{\mathbf{XX}} + \sigma^2 I)^{-1}\mathbf{y} \tag{50}$$

$$= \mathbf{b}(\mathbf{x}, S)^\top \tilde{m}(\mathbf{X}). \tag{51}$$

Analogously, the conditional expectation of the posterior covariance function, i.e., $\mathbb{E}[\tilde{k}(X, X') \mid X_S = \mathbf{x}_S, X'_S = \mathbf{x}'_S]$, can be estimated following the steps above,

$$\mu_{X|X_S=\mathbf{x}_S}^\top \mu_{X'|X'_S=\mathbf{x}'_S} - \mu_{X|X_S=\mathbf{x}_S}^\top k(\cdot, \mathbf{X})(\mathbf{K}_{\mathbf{XX}} + \sigma^2 I)^{-1}k(\mathbf{X}, \cdot)\mu_{X'|X'_S=\mathbf{x}'_S}. \tag{52}$$

After replacing the population conditional mean embedding as their empirical estimates, we can arrive at the solution. $\square$

**Proposition 9** (GP-SHAP)**.** *Let the matrix* $\mathbf{A}$ *be defined as in Theorem 7. The mean and covariance for the multivariate stochastic Shapley values can be estimated as,*

$$\phi\left(\hat{\nu}_f(\mathbf{x}, \cdot)\right) = \mathcal{N}\left(\mathbf{A}\mathbf{B}(\mathbf{x}, [d])^\top \tilde{m}(\mathbf{X}), \mathbf{A}\mathbf{B}(\mathbf{x}, [d])^\top \tilde{\mathbf{K}}_{\mathbf{X}\mathbf{X}}\mathbf{B}(\mathbf{x}, [d])\mathbf{A}^\top\right) \tag{8}$$

*where* $\mathbf{B}(\mathbf{x}, [d]) = [\mathbf{b}(\mathbf{x}, [d]_1), \ldots, \mathbf{b}(\mathbf{x}, [d]_{2^d})]^\top$.

*Proof.* The result follows directly from the previous proposition. Recall $\phi(\hat{\nu}_f(\mathbf{x}, \cdot)) = \mathbf{A}\hat{\mathbf{v}}_{\mathbf{x}}$ for $\hat{\mathbf{v}}_{\mathbf{x}}$ the vector of stochastic payoffs for each coalition. To estimate the mean, we

$$\mathbb{E}[\phi(\hat{\nu}_f(\mathbf{x}, \cdot))] = \mathbf{A}\mathbb{E}[\hat{\mathbf{v}}_{\mathbf{x}}] \tag{53}$$

$$= \mathbf{A}\begin{bmatrix} \hat{m}_\nu(\mathbf{x}, S_1) \\ \vdots \\ \hat{m}_\nu(\mathbf{x}, S_{2^d}) \end{bmatrix} \tag{54}$$

$$= \mathbf{A}\begin{bmatrix} \mathbf{b}(\mathbf{x}, S_1)^\top \tilde{m}(\mathbf{X}) \\ \vdots \\ \mathbf{b}(\mathbf{x}, S_{2^d})^\top \tilde{m}(\mathbf{X}) \end{bmatrix} \tag{55}$$

$$= \mathbf{A}\mathbf{B}(\mathbf{x}, [d])^\top \tilde{m}(\mathbf{X}). \tag{56}$$

Recall $\mathbb{V}[\mathbf{v}_{\mathbf{x}}]_{i,j} = \hat{k}_\nu((\mathbf{x}, S_i), (\mathbf{x}, S_j)) = \mathbf{b}(\mathbf{x}, S_i)^\top \tilde{\mathbf{K}}_{\mathbf{X}\mathbf{X}}\mathbf{b}(\mathbf{x}, S_j)$, the derivation for the covariance matrix then follows analogously as the derivation for the mean,

$$\mathbb{V}[\phi(\hat{\nu}_f(\mathbf{x}, \cdot))] = \mathbf{A}\mathbb{V}[\hat{\mathbf{v}}_{\mathbf{x}}]\mathbf{A}^\top \tag{57}$$

$$= \mathbf{A}\left[\mathbf{b}(\mathbf{x}, S_i)^\top \tilde{\mathbf{K}}_{\mathbf{X}\mathbf{X}}\mathbf{b}(\mathbf{x}, S_j)\right]_{i=1, j=1}^{2^d, 2^d} \mathbf{A}^\top \tag{58}$$

$$= \mathbf{A}\mathbf{B}(\mathbf{x}, [d])^\top \tilde{\mathbf{K}}_{\mathbf{X}\mathbf{X}}\mathbf{B}(\mathbf{x}, [d])\mathbf{A}^\top. \tag{59}$$

$\square$

**Proposition 10** (BayesSHAP [20])**.** *Given the data generation above, the posterior distribution on* $\bar{\phi}$ *and* $\sigma^2$ *follows:*

$$\bar{\phi} \mid \sigma^2, \mathbf{Z}_\ell, f, \mathbf{x}, \mathbf{D} \sim \mathcal{N}(\mathbf{A}_\ell \bar{\mathbf{v}}_{\mathbf{x}}, (\mathbf{Z}_\ell^\top \mathbf{W}_\ell \mathbf{Z}_\ell)^{-1}\sigma^2) \tag{11}$$

$$\sigma^2 \mid \mathbf{Z}_\ell, f, \mathbf{x}, \mathbf{D} \sim \text{Scaled-Inv-}\chi^2\left(\ell_0 + \ell, \frac{\ell_0 \sigma_0^2 + \ell s^2(\bar{\mathbf{v}}_{\mathbf{x}})}{\ell_0 + \ell}\right) \tag{12}$$

*where* $\ell$ *is the number of coalitions* $\mathcal{S} = \{S_j\}_{j=1}^\ell$ *we sample uniformly from* $2^{[d]}$, $\mathbf{Z}_\ell$ *is the binary matrix representing* $\mathcal{S}$, *and* $\mathbf{W}_\ell$ *is the corresponding weight matrix, and* $\mathbf{A}_\ell = (\mathbf{Z}_\ell^\top \mathbf{W}_\ell \mathbf{Z}_\ell)^{-1}\mathbf{Z}_\ell^\top \mathbf{W}_\ell$ *is the WLS matrix,* $\bar{\mathbf{v}}_{\mathbf{x}} = [\bar{\nu}_f(\mathbf{x}, S_1), ..., \bar{\nu}_f(\mathbf{x}, S_\ell)]^\top$ *is the vector of deterministic payoffs, and*

$$s^2(\bar{\mathbf{v}}_{\mathbf{x}}) = \frac{1}{\ell}\left[(\bar{\mathbf{v}}_{\mathbf{x}} - \mathbf{Z}_\ell \mathbf{A}_\ell \bar{\mathbf{v}}_{\mathbf{x}})^\top W_\ell(\bar{\mathbf{v}}_{\mathbf{x}} - \mathbf{Z}_\ell \mathbf{A}_\ell \bar{\mathbf{v}}_{\mathbf{x}}) + (\mathbf{A}_\ell \bar{\mathbf{v}}_{\mathbf{x}})^\top (\mathbf{A}_\ell \bar{\mathbf{v}}_{\mathbf{x}})\right] \tag{13}$$

*measures the average weighted error in the regression and the norm of the mean explanations.*

*Proof.* See Slack et al. [20, Section. 3.1]. $\square$

**Proposition 11** (**BayesGP-SHAP**)**.** *Continuing from Propositions 9 and 10, the posterior distribution of the stochastic Shapley values can be estimated using the Bayesian WLS approach as,*

$$\phi \mid \sigma^2, \mathbf{Z}_\ell, \mathbf{x}, \mathbf{D} \sim \mathcal{N}\left(\mathbf{A}_\ell \mathbf{B}(\mathbf{x}, \mathcal{S}))^\top \tilde{m}(\mathbf{X}), \mathbf{A}_\ell \mathbf{B}(\mathbf{x}, \mathcal{S})^\top \tilde{\mathbf{K}}_{\mathbf{X}\mathbf{X}}\mathbf{B}(\mathbf{x}, \mathcal{S})\mathbf{A}_\ell^\top + (\mathbf{Z}_\ell^\top \mathbf{W}_\ell \mathbf{Z}_\ell)^{-1}\sigma^2\right)$$

*where* $\sigma^2$ *is sampled from* $\sigma^2 \mid \mathbf{Z}_\ell \sim \text{Scaled-Inv-}\chi^2\left(\ell_0 + \ell, \frac{\ell_0 \sigma_0^2 + \ell s^2(\mathbb{E}[\mathbf{v}_{\mathbf{x}}])}{\ell_0 + \ell}\right)$.

*Proof.* We drop the bar notation of $\bar{\phi}$ to unify notations. Given the posterior GP $f \mid \mathbf{D} \sim \mathcal{GP}(\tilde{m}, \tilde{k})$

$$p(\phi \mid \sigma^2, \mathbf{Z}_\ell, \mathbf{x}, \mathbf{D}) = \int p(\phi \mid \sigma^2, \mathbf{Z}_\ell, f, \mathbf{x}, \mathbf{D})p(f \mid \mathbf{D})df \tag{60}$$

Using a standard Gaussian conjugacy procedure, we can derive the variance as the sum of variances from GP-SHAP and BayesSHAP. While it is possible to integrate $p(\sigma^2 \mid \mathbf{Z}_\ell, f, \mathbf{x}, \mathbf{D})$ with respect to the posterior, this leads to a complex scaled mixture of normals that is difficult to model. Instead, we construct a scaled inverse chi-square distribution with $s^2(\mathbb{E}[\mathbf{v_x}])$, which represents the error of the weighted regression with respect to the mean payoffs $\mathbb{E}[\mathbf{v_x}]$. We sample $\sigma^2$ from the following distribution:

$$\sigma^2 \mid \mathbf{Z}_\ell, \mathbf{x}, \mathbf{D} \sim \text{Scale-Inv-}\chi^2\left(\ell_0 + \ell, \frac{\ell_0\sigma_0^2 + \ell s^2(\mathbb{E}[\mathbf{v_x}])}{\ell_0 + \ell}\right). \tag{61}$$

$\square$

## B.4   Proofs for section 4 on predictive explanation and Shapley prior

**Proposition 12** (The Shapley prior over $\phi$). *The prior $f \sim \mathcal{GP}(0, k)$ and the corresponding stochastic game $\nu_f(\mathbf{x}, S) = \mathbb{E}[f(X) \mid X_S = \mathbf{x}_S]$ induce a vector-valued GP prior over the explanation functions $\phi \sim \mathcal{GP}(0, \kappa)$ where $\kappa : \mathcal{X} \times \mathcal{X} \to \mathbb{R}^{d \times d}$ is the matrix-valued covariance kernel*

$$\kappa(\mathbf{x}, \mathbf{x}') = \mathcal{A}(\mathbf{x})^\top \mathcal{A}(\mathbf{x}'), \quad \mathcal{A}(\mathbf{x}) = \Psi(\mathbf{x})\mathbf{A}^\top \tag{14}$$

*where $\Psi(\mathbf{x}) = \left[\mathbb{E}[k(\cdot, X) \mid X_{S_1} = x_{S_1}], \ldots, \mathbb{E}[k(\cdot, X) \mid X_{S_{2d}} = x_{S_{2d}}]\right]$, and the $^\top$ sign refers to taking inner products in the RKHS of $k$.*

*Proof.* The proof is similar to how we proved previous propositions but applied to prior GP $f \sim \mathcal{GP}(0, k)$ instead. If we set,

$$\nu_f(\mathbf{x}, S) = \mathbb{E}[f(X) \mid X_S = \mathbf{x}_S], \tag{62}$$

then $\nu_f$ is a GP on the joint space of data and coalitions with mean 0, and covariance function,

$$\text{cov}\left(\nu_f(\mathbf{x}, S), \nu_f(\mathbf{x}', S')\right) = \mathbb{E}[k(X, X') \mid X_S = \mathbf{x}_S, X'_{S'} = \mathbf{x}'_{S'}] \tag{63}$$

$$= \mu_{X|X_S=\mathbf{x}_S}^\top \mu_{X|X_{S'}=\mathbf{x}'_{S'}}. \tag{64}$$

Since $\phi = \mathbf{A}\mathbf{v_x}$ for $\mathbf{v_x}$ the vector of stochastic payoff from the game induced by the GP prior, the mean stays 0, and the covariance is,

$$\kappa(\mathbf{x}, \mathbf{x}') = \mathbf{A}\left[\mu_{X|X_{S_i}=\mathbf{x}_{S_i}}^\top \mu_{X|X_{S_j}=\mathbf{x}'_{S_j}}\right]_{i=1, j=1}^{2^d, 2^d} \mathbf{A}^\top \tag{65}$$

$$= \mathbf{A}\Psi(\mathbf{x})^\top \Psi(\mathbf{x}')\mathbf{A}^\top \tag{66}$$

$$= \mathcal{A}(\mathbf{x})^\top \mathcal{A}(\mathbf{x}'), \tag{67}$$

therefore we have a matrix-valued covariance kernel $\kappa$ to build a prior over the induced Shapley values. $\square$

**Proposition 13** (Predictive explanations as multi-output GPs). *Given $\mathbf{D}_\phi = \{(\mathbf{x}_i, \phi_i)\}_{i=1}^n = (\mathbf{X}, \mathbf{\Phi_X})$ where $\phi_i \in \mathbb{R}^d$ are the Shapley values computed under predictive model $f$ and $\mathbf{\Phi_X} = [\phi_1, ..., \phi_n]^\top$, the predictive explanations for new data $\mathbf{x}'$ is distributed as,*

$$\phi(\mathbf{x}') \mid \mathbf{D}_\phi \sim \mathcal{N}\left(\tilde{m}_\phi(\mathbf{x}'), \quad \kappa(\mathbf{x}', \mathbf{x}') - \kappa(\mathbf{x}', \mathbf{X})b_\kappa(\mathbf{x}', \mathbf{X})\right) \tag{15}$$

*where $\tilde{m}_\phi(\mathbf{x}') = b_\kappa(\mathbf{x}', \mathbf{X})^\top \text{vec}(\mathbf{\Phi_X})$, $b_\kappa(\mathbf{x}', \mathbf{X}) := (\mathcal{K}_{\mathbf{XX}} + \sigma_\phi^2 I)^{-1}\kappa(\mathbf{X}, \mathbf{x}')$, $\mathcal{K}_{\mathbf{XX}}$ is the gram matrix for kernel $\kappa$ of size $nd \times nd$, $\kappa(\mathbf{x}', \mathbf{X}) = [\kappa(\mathbf{x}', \mathbf{x}_1), \ldots, \kappa(\mathbf{x}', \mathbf{x}_n)]$ is of size $d \times nd$ and $\sigma_\phi^2$ is the noise parameter for regression.*

*Proof.* Follows from standard vector-valued Gaussian process regression results. See Alvarez et al. [51] for a detailed discussion on regression with matrix-valued kernels. $\square$

**Proposition 14** (Posterior mean as Shapley values for payoff vector $\tilde{\mathbf{v}}_{\mathbf{x}'}$). *The posterior mean $\tilde{m}_\phi(\mathbf{x}')$ corresponds to Shapley values for the payoff vector $\tilde{\mathbf{v}}_{\mathbf{x}'}$, i.e., $\tilde{m}_\phi(\mathbf{x}') = \mathbf{A}\tilde{\mathbf{v}}_{\mathbf{x}'}$, where $\tilde{\mathbf{v}}_{\mathbf{x}'} = \sum_{i=1}^n \Psi(\mathbf{x}')^\top \Psi(\mathbf{x}_i)\mathbf{A}^\top \alpha_i$ and $\alpha_i \in \mathbb{R}^d$ is the $[i, ..., i + (d-1)]$ subvector of $(\mathcal{K}_{\mathbf{XX}} + \sigma_\phi^2 I)^{-1} \text{vec}(\mathbf{\Phi_X})$.*

*Proof.* There are two ways to see this. First is by brute force and rearranging the terms in the posterior mean expression. The other is to leverage the vector-valued representer theorem [52] and write the posterior mean as,

$$\tilde{m}_\phi(\mathbf{x}') = \sum_{i=1}^{n} \mathcal{A}(\mathbf{x}')^\top \mathcal{A}(\mathbf{x}_i)\alpha_i, \quad \alpha_i \in \mathbb{R}^d \tag{68}$$

$$= \sum_{i=1}^{n} \mathbf{A}\Psi(\mathbf{x}')^\top \Psi(\mathbf{x}_i)\mathbf{A}^\top \alpha_i \tag{69}$$

$$= \mathbf{A}\left(\sum_{i=1}^{n} \Psi(\mathbf{x}')^\top \Psi(\mathbf{x}_i)\mathbf{A}^\top \alpha_i\right) \tag{70}$$

$$= \mathbf{A}\tilde{\mathbf{v}}_{\mathbf{x}'} \tag{71}$$

after some linear algebra exercises, we can see that $\alpha_i$ is the $[i : i + (d - 1)]$ sub-vector of $(\mathscr{K}_{\mathbf{XX}} + \sigma_\phi^2 I)^{-1} \text{vec}(\mathbf{\Phi_X})$ ∎

## C  Implementation details and further illustrations.

All illustrations are run locally on a MacbookPro 2021 with Apple M1 pro chip.

### C.1  Ablation study on different notions of uncertainties captured

To demonstrate the difference between the uncertainties captured by GP-SHAP, BayesSHAP, and BayesGP-SHAP, we utilise the California housing dataset [41]. This dataset was derived from the 1990 U.S. census, each observation represent a census block group. A block group is the smallest geographical unit for which the U.S. Census Bureau publishes sample data (a block group typically has a population of 600 to 3,000 people). The dataset includes 20640 instances with 8 numerical features measuring the following:

- **MedInc:** Median income in block group
- **HouseAge:** Median house age in block group
- **AveRooms:** Average number of rooms per household
- **AveBedrms:** Average number of bedrooms per household
- **Population:** Block group population
- **AveOccup:** Average number of houehold members
- **Latitude:** Block group latitude
- **Longitude:** Block group longitude

The target variable is the median house value for California districts, expressed in hundreds of thousands of dollars. In the following, we train a GP model and extract explanations using GP-SHAP, BayesSHAP, and BayesGP-SHAP, for 4 different configurations:

1. trained on 25% of data, estimate the Shapley values using 50% of coalitions.
2. trained on 25% of data, estimate the Shapley values using 100% of coalitions.
3. trained on 100% of data, estimate the Shapley values using 50% of coalitions.
4. trained on 100% of data, estimate the Shapley values using 100% of coalitions.

To fit the GP model, we employ a sparse Variational GP approach with 200 learnable inducing point locations. The evidence lower bound is optimized using batch gradient descent with a batch size of 64, a learning rate of 0.01, and 100 iterations. The RBF kernel with learnable bandwidths initialized using the median heuristic approach is used for the sparse GP. The inducing locations are initialized using a standard clustering approach to obtain a representative set of inducing points.

After training the model, we reuse the learned inducing points and kernel bandwidths for the explanation algorithms. The explanations are obtained using the procedure described in Algorithm 1 of our work.

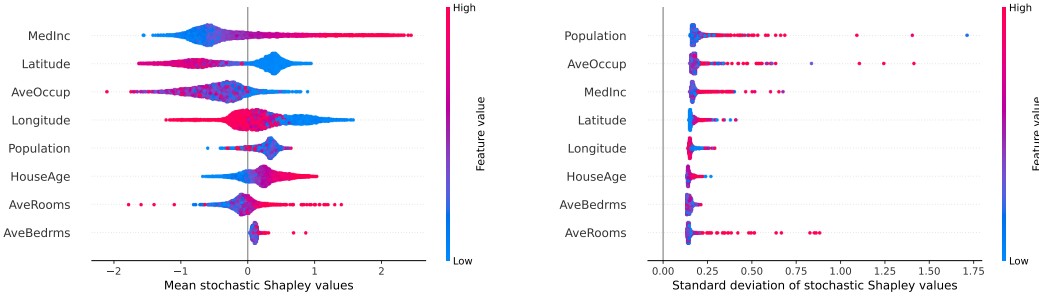

(a) Mean of each stochastic explanations  (b) Standard deviation of each explanations

Figure 4: We plot the beeswarm plot of the mean and standard deviations of each stochastic explanations from BayesGP-SHAP fitted on the housing dataset. The features are ranked according to the distance span by the largest and smallest mean (std) stochastic Shapley values.

In Figure 1 of our paper, we present the stochastic Shapley values for the 11th observation, computed using the three explanation algorithms. The plot includes the 95% credible interval to visualize the uncertainties associated with the explanations.

**Further illustration:** In Figure 4, we plot the beeswarm plot on the mean and standard deviation of each stochastic explanations respectively. We color the point based on the relative size of the feature value compared to the rest. We see that in Figure 4a, which plotted the mean stochastic shapley values for each observation, the relationship between most features' explanation to the target variable is quite linear. For example, the higher the median income (**MedInc**), the more positive those feature contribute to predicting the respective median house value. On the other hand, Figure 4b illustrated the standard deviation of each stochastic explanations. In general, we see that the larger the feature values are, the more uncertain the explanation becomes. Nonetheless, we see that the feature contributing the most, defined as the feature having largest distance spanned by their most positive and most negative mean stochastic Shapley values, does not necessarily have the largest variation respectively.

## C.2 Exploratory analysis of the stochastic explanations

For this illustration, we utilise the breast cancer dataset [42], containing 569 patients with 30 numeric features. They are computed from a digitized image of a fine needle aspirate (FNA) of a breast mass and describe characteristics of the cell nuclei present in the image:

- radius (mean of distances from center to points on the perimeter)
- texture (standard deviation of gray-scale values)
- perimeter
- area
- smoothness (local variation in radius lengths)
- compactness ($\frac{\text{perimeter}^2}{\text{area} - 1}$)
- concavity (severity of concave portions of the contour)
- concave points (number of concave portions of the contour)
- symmetry
- fractal dimension ("coastline approximation" - 1)

The goal is to predict whether a tumour is malignant or benign. We first fit a GP model with RBF kernel using again the sparse Variational GP formulation with $200$ learnable inducing locations. We initialise the inducing points using standard clustering techniques on the data. The evidence lower bound objective is optimised with a learning rate of $1e^{-4}$ and $1000$ iterations using batch gradient descent of batch size $64$. To obtain the explanations, we run the BayesGP-SHAP algorithm with $2^{16}$

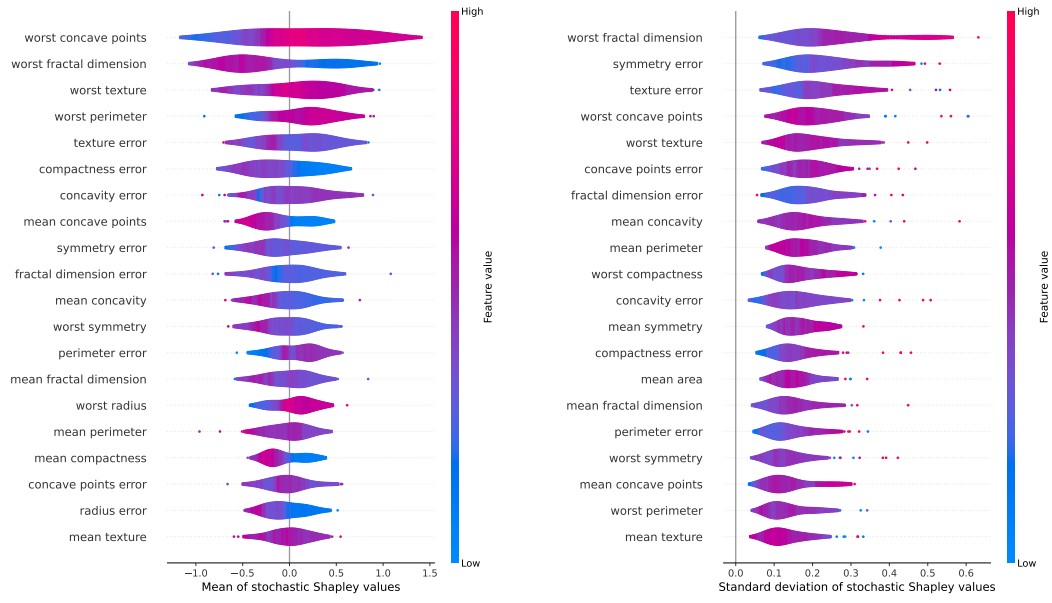

(a) Mean of each stochastic explanations      (b) Standard deviation of each explanations

Figure 5: We plot the violin plot of the mean and standard deviations of each stochastic explanations from BayesGP-SHAP fitted on the breast cancer. The features are ranked according to the distance span by the largest and smallest mean (std) stochastic Shapley values.

number of coalitions. We do not compare GP-SHAP and BayesSHAP here because the BayesSHAP uncertainties have shrunk to almost zero, i.e., the mean standard deviations from the BayesSHAP uncertainties across all features and data is $0.0002$. This reconfirms the fact from Slack et al. [20] that as we increase the sample size the estimation error goes to zero, thus the uncertainties from BayesSHAP goes to zero as well. On the other hand, GP-SHAP uncertainties still remain valid because it represents the GP predictive uncertainties, which do not shrink to zero as we increase the number of coalitions we use to esitmate the SVs.

**Further illustrations:** In Figure 5, we plot two violin plots to illustrate the relationship between mean and standard deviation of the stochastic values with respect to the size of the original feature. We see that the feature "worst fractal dimension" are the second most influential feature in terms of mean stochastic explanations and also the feature that has highest uncertainty around its explanations. In comparison with the housing prediction problem illustrated in Figure 4, the higher the feature value doesn't necessary give higher uncertainty around its explanation.

### C.3 Predictive explanations

For this illustration, we utilise the Diabetes dataset [47] with $442$ patient data and $10$ numeric features measuring the following:

- age: age in years
- sex
- bmi: body mass index
- bp: average blood presuure
- s1: total serum cholesterol
- s2: low-density lipoproteins
- s3: high-density lipoproteins
- s4: total cholesterol
- s5: Log of serum triglycerides level

- s6: blood sugar level

The experiment is to assess the effectiveness of the Shapley prior we proposed in predicting explanations estimated using SHAP algorithms for general models, including GP-SHAP, TreeSHAP, and DeepSHAP. We use the implementation of TreeSHAP and DeepSHAP from the **shap** package [2].

While algorithms such FastSHAP [22] also learn a vector-valued function that returns explanations given instances, the algorithm require access to the underlying model $f$ during training while ours required previously computed explanations. Due to this importance difference in the problem setup, we do not compare the two algorithm.

We first generate three sets of explanations to set up three regression problems:

1. Fit a Gaussian process model and then run GP-SHAP to obtain explanations.

2. Fit a random forest model and then run TreeSHAP to obtain explanations.

3. Fit a neural network model and then run DeepSHAP to obtain explanations.

After obtaining explanations as groundtruths for this experiment, we randomly divide $70\%$ of them as training data and $30\%$ of them as testing data. We then do the following,

1. We fit a multi-output GP using the proposed Shapley prior on the training data and predict the explanations of the unseen test data.

2. We fit a multi-output random forest model on the training data and predict the explanations of the unseen test data.

3. We fit a multi-output neural network model on the training data and predict the explanations of the unseen test data.

We repeat this experiment 10 times using different seeds and compute the RMSE between the predicted and groundtruths explanations. The results are then plotted in Figure 3.

### C.4 Further ablation study: Impact of increased posterior prediction uncertainty on explanation uncertainties

In this ablation study, we aim to examine the effect of increasing the uncertainty in posterior predictions on the corresponding uncertainty in stochastic Shapley values. To demonstrate this, we utilize the diabetic dataset [47] and split the data based on recorded sex. We train our GP model on the male data and employ BayesGP-SHAP to explain the prediction results for both the male training data and the female testing data. We adopt this split because we expect the biological characteristics between males and females to be distinct enough to treat the female data as out-of-sample data, thereby naturally resulting in increased predictive uncertainty for the female data. To further amplify this uncertainty, we multiply each instance in the female testing data by distortion factors of two and three, respectively, and assess the corresponding uncertainties in the explanations.

We begin by illustrating the relationship between the out-of-sampleness of the data and the corresponding increase in predictive posterior uncertainties. This is depicted in Figure 6a, where we observe that as the data becomes more out-of-sample, the predictive uncertainties consistently rise. Even at distortion level 1, which represents the original female data, we can already observe increased uncertainties compared to the uncertainties derived from male data prediction.

Furthermore, these increased uncertainties in the predictive posterior are reflected in the associated feature explanations. This is evident in Figure 6b, where we visualize the uncertainties associated with the feature explanations. For instance, the green bars representing the average uncertainties in explaining female data with no distortion are consistently larger than the red bars, which represent the average uncertainties of male data explanations. This observation aligns with the higher predictive uncertainties observed in Figure 6a for the female data compared to the male training data.

It is worth noting that the uncertainty for the feature "sex" remains consistently close to zero. This is because the feature "sex" is constant within both the female and male datasets. As a result, it acts as the null player in each dataset and obtains an almost Dirac zero as its stochastic Shapley value.

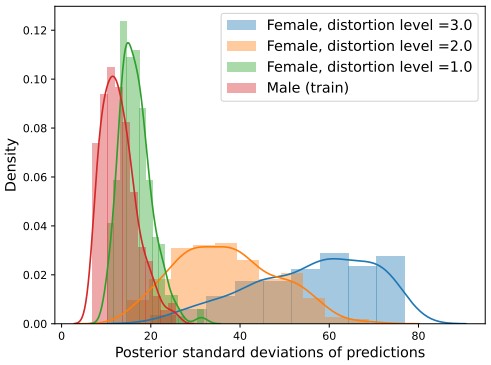

(a) Predictive posterior standard deviation

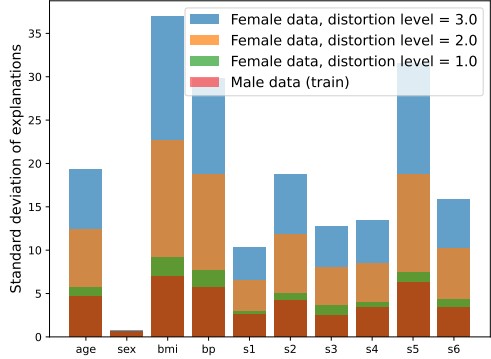

(b) Mean of standard deviations of explanations

Figure 6: Ablation study: (left) We begin by training a Gaussian Process (GP) model on the male data. We then make predictions using this trained model on both the male data and out-of-sample female data. To assess the impact of increasing posterior uncertainties, we multiply the female data by distortion levels of 1.0, 2.0, and 3.0. We visualize the results by plotting the density plot of the standard deviations obtained from the predictive posterior distributions. (right) Next, we focus on analyzing the average standard deviations of explanations per feature from the male and female data, considering different distortion levels. We observe that as we progressively increase the posterior uncertainties in the sample, these uncertainties are reflected in the uncertainties of the explanations provided.

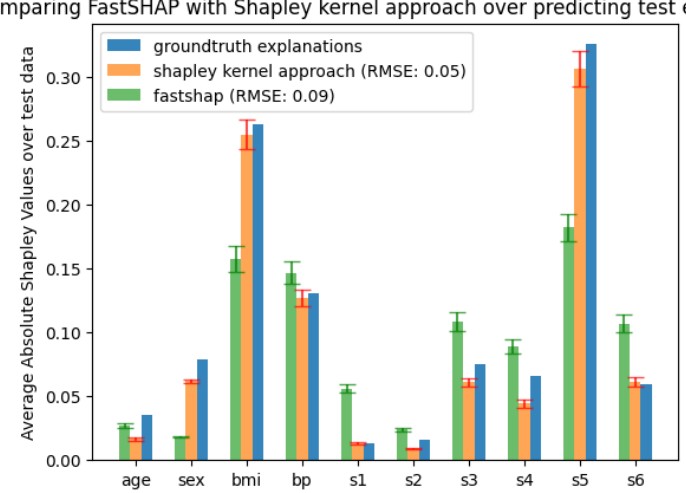

Figure 7: Comparing the predicted Shapley values with groundtruth from GP regression with Shapley kernel versus fastSHAP on the diabetic dataset.

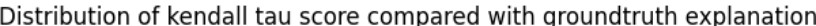

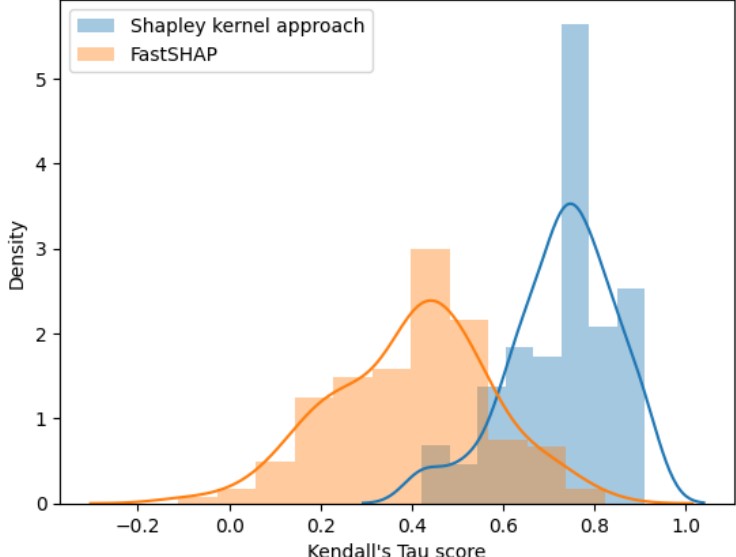

Figure 8: Comparing the Kendall's Tau score between the feature importance ranking from GP regression with Shapley kernel and FastSHAP with groundtruth feature importance ranking on the diabetic dataset.

## D Comparison with FastSHAP [39]

We ran a predictive explanation experiment (see Figure 8 and 7 ) using the diabetic dataset from UCI, comparing both our method and FastSHAP. In particular, we use 70% of the data as training, i.e. we first compute the Shapley values for training data and treat them as labels, and predict the explanation for the remaining data. We evaluated performance using both RMSE and Kendall's tau distance (the closer to 1 the better). Notably, Kendall's tau distance assesses the consistency in feature importance ranking between predicted explanations and the true explanations. In the experiments, our method got RMSE 0.05, while FastSHAP got 0.09. In terms of Kendall's tau, our method got an average around 0.7 while FastSHAP got around 0.4, meaning the feature importance ranking recovered from our method is more aligned with the groundtruth SVs than FastSHAP. Our findings indicate that our method outperformed FastSHAP in this experiment.

Note however that interpretation of these comparisons should be nuanced since the two techniques have different usecases: FastSHAP uses knowledge of the model, not the previously computed SVs, whereas GPSHAP uses previously computed SVs and is agnostic to the type of model they came from.

