# OpenReview forum: "Explaining the Uncertain: Stochastic Shapley Values for Gaussian Process Models"
_NeurIPS.cc/2023/Conference — NeurIPS 2023 spotlight_

### Official Review · Reviewer_uH3p · 2023-06-28

**Soundness:** 4 excellent
**Presentation:** 4 excellent
**Contribution:** 3 good
**Rating:** 7
**Confidence:** 4

**Summary:**

The work proposes a new kind of Shapley value (SV) based explanation method. The new methods GP-SHAP and BayesGP-SHAP are like TreeSHAP, LinearSHAP or RKHS-SHAP model-specific SV methods for Gaussian Process (GP) models. The authors further propose another approach of utilizing GP models to amortize (i.e learn) SVs for a model and data point in a predictive task (like FastSHAP). The authors name this the Shapley prior. The two main contributions (model-specific) SV method and the Shapley prior is illustrated in experiments with well-established benchmark datasets.

**Strengths:**

The paper has many strengths. The work is theoretically well-formulated and motivated. It combines uncertainty research with XAI research to enrich the quality of explanations. This is very important research. The contribution is well formulated and well motivated (formally and verbally). The experimental analysis of the model-specific GP-SHAP and BayesGP-SHAP is convincing. While the work is very theoretical in nature, it is still very pleasant to read, which is hard to achieve, since bridging the gap between two streams of research is not easy.
The work adheres perfectly to the style guidelines, includes a well-rounded appendix with code and a small readme file for easier reproducibility. The proofs are to the most part understandable.

**Weaknesses:**

The main weaknesses I identified are elaborated below:

1. The predictive (FastSHAP-like) approach to learning SVs with GP models and the Shapley prior feels a bit tagged on and underdeveloped. In lines 256-258 the argument is made that GP models are trained only with x (instance) and y (SV for instance) pairs and the model is not required to train. This is posed as a positive side of this Shapley prior approach. This, of course, is correct that you only need the SV to train, however, doesn't this make the GP models less effective in comparison to, for instance, FastSHAP which is trained on this Shapley loss function (i.e. MSE over model output of coalition with feature i than without feature i). The main benefit I see is that you now have a GP model rather than a big black-box NN like FastSHAP proposes. The GP models, therein, are more explainable which is a desideratum in my opinion for an explanation system (this "explaining a model with another model that needs to be explained in itself" can be spun ad absurdum). However, the paper is missing such an experimental comparison of the Shapley prior with FastSHAP or an in-depth discussion of the benefits in my opinion. I want to reiterate that this is not a big problem, however, it is left hanging.

2. The work has some accessibility issues from a ML perspective. Some concepts are hard to follow even with a good background in SV and statistics. I recon that ML readers might have a hard time understanding the contribution well because they already have problems with the problem definitions (beginning of 2). If space allows I'd suggest to certainly enrich the definition section (lines 81-92) with some explanations.

3. The connection to uncertainty research could be a bit stronger. I.e. it would be nice to include a more extensive discussion of what parts of uncertainty are captured in the confidence intervals (i.e. aleatoric or epistemic)?

Minor things:
- Line 11 ... "extensive illustrations" sounds a bit counterintuitive consider changing this
- Write out the main contribution again in the introduction in a seperate paragraph that is easily digestible for ML engineers
- Line 134 "rvs" is introduced but (I think) not used. Even if it is used consider removing this abbreviation
- Lines 235-240: I would like to see a more extensive discussion here.

**Questions:**

1. How does your approach compare to FastSHAP in terms of predictive performance? What's the tradeoff in terms of predictive performance and explainability?

2. Can you say anything about what parts of uncertainty are captured in the explanations uncertainty (i.e. aleatoric or epistemic)?

**Limitations:**

The work includes a limitation section discussing (a) limitations and (b) future work. However, the section is quite short and focuses solely on future work and avenues to explore (e.g. extension to Bayesian NN). A more detailed discussion of the limitations of this particular result would be beneficial.

---

> ### Author Rebuttal · Authors · 2023-08-07
>
> We thank the reviewer for their constructive feedbacks and we will answer their questions below:
>
> ## 1. How does your approach compare to FastSHAP in terms of predictive performance? What's the tradeoff in terms of predictive performance and explainability?
> - To answer this, we have ran a predictive explanation experiment (please see attached pdf in global response) using the diabetic dataset from UCI, comparing both our method and FastSHAP. The results are included in the one-page pdf submission. In particular, we use 70% of the data as training, i.e. we first compute the Shapley values for training data and treat them as labels, and predict the explanation for the remaining data. We evaluated performance using both RMSE and Kendall’s tau distance (the closer to 1 the better). Notably, Kendall’s tau distance assesses the consistency in feature importance ranking between predicted explanations and the true explanations. In the experiments, our method got RMSE 0.05, while FastSHAP got 0.09. In terms of Kendall's tau, our method got an average around 0.7 while FastSHAP got around 0.4, meaning the feature importance ranking recovered from our method is more aligned with the groundtruth SVs than FastSHAP. Our findings indicate that our method outperformed FastSHAP in this experiment.
> - **Note however that interpretation of these comparisons should be nuanced since the two techniques have different usecases: FastSHAP uses knowledge of the model, not the previously computed SVs, whereas GPSHAP uses previously computed SVs and is agnostic to the type of model they came from.**
> ---
> ## 2. Can you say anything about what parts of uncertainty are captured in the explanations’ uncertainty (i.e. aleatoric or epistemic)?
> - Both GPSHAP and BayesSHAP are capturing epistemic uncertainties, but from different sources. GPSHAP captures epistemic uncertainty from fitting the GP function f, while BayesSHAP captures epistemic uncertainty arising when estimating the Shapley values using a Bayesian weighted least square formulation (for a fixed function f). BayesGPSHAP then captures both uncertainties.

---

> > ### Comment · Reviewer_uH3p · 2023-08-16
> > **Response to the authors**
> >
> > Thank you for replying to my questions and your work. I feel they have been adequately addressed. I will keep my score unchanged.  I think the work would benefit from an addition of a short discussion on what part of uncertainty is captured in the main body of the work. Further, the additional experiments and more detailed comparison to FastSHAP would be a welcome addition to the appendix.

---

### Official Review · Reviewer_hgou · 2023-07-04

**Soundness:** 4 excellent
**Presentation:** 3 good
**Contribution:** 3 good
**Rating:** 6
**Confidence:** 3

**Summary:**

The authors consider Shapley values computed from games $\nu$ modelled with a Gaussian process. Since the game is stochastic, the Shapley values are now stochastic, and the stochasticity reflects epistemic uncertainty in the explanations. This is unlike previous methods like BayesSHAP, where the uncertainty is estimation error. The approach of the authors, called GP-SHAP, can be used to explain predictions of Gaussian processes. It can also be used to construct a prior over Shapley values, which in turn can be used to perform regression on Shapley values. The authors demonstrate the utility of their approach is three experiments.

**Strengths:**

I would like to thank the authors for their submission. I think this is interesting work.

## Strengths

* The paper is well written and mostly clear. The presentation is reasonably good.

* Pushing a GP through the value allocation computation to explain stochastic predictions seems like a very sensible thing to do.

* Doing the above to define a prior over value allocations and then using this prior for regression of Shapley values is, to the best of my knowledge, novel. I have not seen this before. I should, however, say that I'm not at all familiar with this part of the literature.

* The stochasticity in explanations by GP-SHAP represents epistemic uncertainty in the explanation, whereas the stochasticity in explanations by BayesSHAP is due to estimation error. I think this is an interesting finding.

* I have either skimmed through or looked in more detail at the proofs of the mathematical statements. Apart from some clarity issues (see below), I think everything generally checks out.

* The experiments nicely illustrate the benefits of GP-SHAP.


## General Remarks

* In Proposition 11, I was initially very confused by that the expectation of $v_x$ appears. After reading the proof, it became clear that this is an approximation that simplifies the computation, which should actually integrate over $f$. I think this can be better explained in the statement of Propostion 11.

* In Proposition 12, if I'm not mistaken, the transpose in the expression for $\kappa$ refers to taking inner products in the RKHS of $k$. This is _definitely_ not clear at all and should be clarified!

* In Figure 1, how does BayesSHAP produce its explanations? It is applied to the predictive mean of the GP prediction?

**Weaknesses:**

## Weaknesses

* I think that providing a proof for Theorem 4 is unnecessarily complicating the exposition, as the Shapley's original proof for the deterministic case can be applied to immediately prove Theorem 4: Let $(\Omega, \mathcal{F}, \mathbb{P})$ be the underlying probability space. For any fixed sample $\omega \in \Omega$, $\nu$ is d-game (using the authors' language). Therefore, according to Shapley's original proof, for that $\omega \in \Omega$, it can be written in the form of (1). Since (1) holds for all $\omega \in \Omega$, it obviously holds as an equality in terms of random variables.

* Generally, I think that the exposition takes a long time to arrive at the key idea of the paper: model $\nu$ with a GP, and push it through the value allocation computation in (1). I think that the exposition would be improved by having Section 3 starting out with immediately explaning this key idea. A paper is not supposed to be a novel: please signpost important ideas and conclusions as much as possible.

* In lines 326-339, you highlight the difference between the mean of absolute SSVs and the absolute values of mean SSVs, and state that this gives a different ordering for which feature is most influential. However, you do not analyse whether this different ordering is better or worse, which means that it is not clear whether, in this case, GP-SHAP produced a better ordering or not.

* In lines 340-346, you produce a local explanation graphical model. However, you do not analyse whether this graphical model is reasonable, which means that the reader is not sure whether GP-SHAP did something sensible or not.

* Building a GP prior over Shapley values to do regression is an interesting idea. However, the presentation would be more convincing if you were to point out some existing applications that would actually want to do regression of Shapley values.


**Questions:**

## Conclusion

I think this is a solid submission without any major shortcomings, which is why I am giving an accept.

If at all possible, I would like to see the following edits in a revision of the authors:

* If I'm right that Theorem 4 follows immediately from the deterministic case, I would really just do that and omit the current proof.

* Please start out Section 3 by explaning what you are working towards: pushing a GP through the value allocation computation

* Please clarify the statement of Proposition 11 (see above).

* Please clarify the transpose in Proposition 12 (see above).

* In Figure 2.(b), please analyse the difference in ordering between the mean of absolute SSVs and the absolute values of SSVs and conclude which of the two orderings is more sensible.

* In Figure 2.(d), please argue whether the graphical model is sensible or not.

EDIT

I have read the authors' rebuttal and the other reviews and remain at my current assessment.

**Limitations:**

See above.

---

> ### Author Rebuttal · Authors · 2023-08-07
>
> We thank the reviewer for their in-depth suggestions. We would like to address their concerns below:
>
> ### Proof of Theorem 4:
> We agree with the reviewer that using the d-game to establish stochastic Shapley values as a push forward measure is easier to understand. In fact, this approach was the one adopted by Ma et al. 2008, where they first proposed a potential solution, then verified its alignment with the three (stochastic) axioms and its uniqueness—a top-down proof strategy. For our study, as explained in lines 561-565, we aimed to offer a contrasting perspective that hasn't appeared in the literature, mirroring Shapley's original bottom-up derivation. This means we began with the axioms and subsequently determined the unique solution. Recognising the potential for confusion, we have incorporated the following explanation in the main text under theorem 4 to clarify our approach:
> - **``We emphasise that this result has been proven in Ma et al. using a top-down approach, i.e. starting with Equation (1) and then verifying that it satisfies the stochastic axioms and uniqueness. In the appendix, we offer a contrasting perspective where we mirror Shapley's original bottom-up derivation, i.e. we began with the stochastic axioms and subsequently determined the unique solution''**
>
> [Ma et al. 2008] Ying Ma, Zuofeng Gao, Wei Li, Ning Jiang, Lei Guo, et al. The shapley value for stochastic
> cooperative game. Modern Applied Science, 2(4):1–76, 2008.
>
> ---
>
> ### Clarity
> We have incorporated the reviewer’s suggestion and included a high level description of key ideas in the beginning of section 3. Both propositions 11 and 12 are now clarified as well. Particularly, we wrote:
> - High level description: **"This section is dedicated to introducing GP-SHAP, a method which efficiently computes stochastic Shapley values for stochastic games where the value function is computed using conditional expectations of Gaussian processes."**
> - Proposition 11: **"We note that in the above proposition, instead of integrating $p(\sigma^2\mid \mathbf{Z}_\ell, f, \mathbf{x}, \mathbf{D})$ with respect to the posterior GP, which leads to a complex scaled mixture of Gaussians, we simplify the computation and instead construct a scaled inverse chi-square distribution with $s^2(\mathbb{E}[\mathbf{v}_x])$ instead, which represents the error of the weighted regression with respect to the mean payoffs."**
> - Proposition 12: We will add this sentence to the end of prop 12, **".. and the inner product is taken over the RKHS of $k$"**
>
> ---
>
> ### Global explanation
> We believe that it's a more principled approach to measure global importance by computing the mean of absolute SSVs rather than taking the absolute values of the mean SSVs, which neglects uncertainty in SSVs. Practitioners employing GP-based models would likely prioritise leveraging quantified uncertainty in their explanations. By simply using the absolute values of mean SSVs, they neglect to incorporate this valuable uncertainty information. We have added the following discussion in the main paper to make the exposition clearer:
> - **"We believe the former (mean of absolute SSV) approach is more appropriate than the latter (absolute of mean SSVs) as practitioners employing Gaussian processes models would want their global explanations to take the quantified uncertainty into account."**
>
> ---
> ### Graphical model:
> We emphasise that, analogously to the Shapley values as explanation tool, the graphical model depicting dependence structure of stochastic Shapley values is a tool for explaining the fitted GP model, not the true underlying data generating processes, so whether or not this graphical model is sensible will depend on whether the fitted GP model is itself sensible. Nonetheless, Gaussian graphical model is a faithful representation of the multivariate normal distribution of the Shapley values that is induced by the fitted GP model, and such representation of multivariate normal distributions is standard in the literature.
>
> ---
>
> We thank the reviewer again for their suggestions on the revision. As we have answered and clarified the reviewer's comments, we hope that the reviewer could revise their score from a weak accept to accept, in light of our responses and other reviewers' positive feedback. Thank you.

---

> > ### Comment · Reviewer_hgou · 2023-08-10
> >
> > Thank you for your reply to my review. After reading the other reviews and again assessing the strengths and weaknesses, I have decided to maintain my current score.

---

### Official Review · Reviewer_FXFV · 2023-07-06

**Soundness:** 4 excellent
**Presentation:** 3 good
**Contribution:** 4 excellent
**Rating:** 8
**Confidence:** 3

**Summary:**

This works introduces the stochastic Shapley values to the explainable AI community. It formulates the stochastic Shapley values for GP models with a closed-form expression for the covariance matrix. Then, it proposes BayesGP-SHAP, which incorporates both sources (of uncertainty from the GP posterior and of the Shapley value estimation). Lastly, it constructs an inference model by proposing a GP prior to the Shapley value to solve the problem of predictive explanation.


**Strengths:**

I enjoy reading the paper as it contains several major strengths:
+ The motivation of predictive uncertainty in trustworthy ML models is convincing, which paves the way to the proposed GP-SHAP and BayesGP-SHAP.
+ The literature review is sufficient to highlight the novelty of the approach.
+ It introduces concepts of stochastic cooperative games and stochastic Shapley values to the explainable AI community.
+ GP explanations not only satisfy favorable axioms to standard Shapley values but are also able to measure explanation uncertainties and determine statistical dependencies between explanations.
+ The proposed BayesGP-SHAP integrates both sources of uncertainty from the GP posterior and the estimation of the Shapley value.
+ It proposes the Shapley prior to dealing with the problem of predictive explanation, which is a nice Bayesian treatment to the predictive explanation problem.
+ In the experiments, the paper introduces several exploratory analysis methods for practitioners to understand stochastic explanations.


**Weaknesses:**

The paper may elaborate on other use cases of predictive explanations. Anyways, I find the Bayesian optimization application in the conclusion interesting.

There are very minor issues in the references as some words are not capitalized, e.g., Shapley.


**Questions:**

I do not have any questions for the authors.


**Limitations:**

The authors have adequately stated the limitations in the appendix.

---

> ### Author Rebuttal · Authors · 2023-08-07
>
> We thank the reviewer for their positive feedback and pointing out the typos in the references, we will amend them accordingly in the camera ready version.

---

### Official Review · Reviewer_1zjr · 2023-07-07

**Soundness:** 4 excellent
**Presentation:** 4 excellent
**Contribution:** 4 excellent
**Rating:** 7
**Confidence:** 3

**Summary:**

The paper introduces a novel approach for explaining Gaussian processes (GPs) by utilizing the analytical covariance structure present in GPs based on extending the concept of Shapley values to stochastic cooperative games, resulting in explanations that are random variables. The GP explanations generated using this approach satisfy favorable axioms similar to standard Shapley values and possess a tractable covariance function across features and data observations. The proposed approach, GP-SHAP, and its variant BayesGP-SHAP, are shown to be effective in providing explanations for Gaussian process models.

**Strengths:**

The paper presented a novel and principled approach for explaining GP model using stochastic Shapley values. The proposed approach is sound and thoroughly evaluated. It also fills the void that there is no model-specific explanation method for GP.

**Weaknesses:**

The paper should also discuss the limit of using Shapley value in explanation in general.

**Questions:**

The font in Figure 2 is too small to read.

**Limitations:**

Shapley-value based explanations have some limitations -- two references are provided as follows.
* Shapley Residuals: Quantifying the limits of the Shapley value for explanations (https://proceedings.neurips.cc/paper/2021/file/dfc6aa246e88ab3e32caeaaecf433550-Paper.pdf)
* Problems with Shapley-value-based explanations as feature importance measures (https://arxiv.org/pdf/2002.11097.pdf)

These limits are inherited in this proposed method.

---

> ### Author Rebuttal · Authors · 2023-08-07
>
> We thank the reviewer for sharing literature pointing out the limitations of Shapley-value based explanations. We have included a discussion and included the references in the discussion section in the camera ready version. We will also enlarge figure 2 in the camera ready version.

---

### Author Rebuttal · Authors · 2023-08-09

We include further experiments to compare FastSHAP with GP regression using the Shapley prior in light of reviewer uH3p's comments.

---

### Decision · Program_Chairs · 2023-09-21

**Decision:**

Accept (spotlight)

**Comment:**

There is a clear consensus among the reviewers that this paper makes an interesting and substantial contribution. From the beginning, they have all been on the positive side, although two reviewers also raised a couple of concerns and made suggestions for improvement. Nevertheless, after the rebuttal, all reviewers are still in favor of acceptance, and so am I as well.